# MYC Deregulation and PTEN Loss Model Tumor and Stromal Heterogeneity of Aggressive Triple-Negative Breast Cancer

Zinab O. Doha[1,2,13], Xiaoyan Wang[1,13], Nicholas L. Calistri [3], Jennifer Eng [1,3,4], Colin J. Daniel[1], Luke Ternes[3], Eun Na Kim[3], Carl Pelz[1,5], Michael Munks[5,6], Courtney Betts[7], Sunjong Kwon[3,4], Elmar Bucher[3,4], Xi Li[8], Trent Waugh[1], Zuzana Tatarova[3,4], Dylan Blumberg[3,4], Aaron Ko[6], Nell Kirchberger[7], Jennifer A. Pietenpol[9,10], Melinda E. Sanders[10,11], Ellen M. Langer [1], Mu-Shui Dai [1], Gordon Mills [5,8,12], Koei Chin [3,4,5,12], Young Hwan Chang [3,12], Lisa M. Coussens [5,7,12], Joe W. Gray [3,4,5,12], Laura M. Heiser [3,4,12] & Rosalie C. Sears [1,5,12] ✉

Triple-negative breast cancer (TNBC) patients have a poor prognosis and few treatment options. Mouse models of TNBC are important for development of new therapies, however, few mouse models represent the complexity of TNBC. Here, we develop a female TNBC murine model by mimicking two common TNBC mutations with high co-occurrence: amplification of the oncogene *MYC* and deletion of the tumor suppressor *PTEN*. This Myc;Ptenfl model develops heterogeneous triple-negative mammary tumors that display histological and molecular features commonly found in human TNBC. Our research involves deep molecular and spatial analyses on Myc;Ptenfl tumors including bulk and single-cell RNA-sequencing, and multiplex tissue-imaging. Through comparison with human TNBC, we demonstrate that this genetic mouse model develops mammary tumors with differential survival and therapeutic responses that closely resemble the inter- and intra-tumoral and microenvironmental heterogeneity of human TNBC, providing a pre-clinical tool for assessing the spectrum of patient TNBC biology and drug response.

TNBC represents 10–15% of all breast carcinomas with poor clinical outcomes and greater mortality compared with non-TNBC[1–4]. With current standard therapies, the median OS for the disease is 10.2 months, with a 5-year survival rate of ~65% for patients with regional tumors and 11% for those with the disease spread to distant organs[5,6]. In addition to the aggressive nature of TNBC, the limited, targeted therapy options and lack of sensitivity to endocrine agents contribute to significantly shorter disease-free and overall survival (OS)[1,4]. Although therapeutic options such as targeting immune checkpoints and PARP inhibitors are changing the landscape, TNBC is still currently the worst outcome of breast cancer. Together this indicates an urgent need for a

deeper understanding of this disease that can lead to the identification of more effective therapeutic strategies.

Approximately 70% of triple-negative tumors are molecularly classified as basal-like[7–9], a subtype characterized by aggressive phenotypes with higher rates of proliferation, poor differentiation, and increased metastatic capability[10]. More recent studies have defined four TNBC subtypes: luminal androgen receptor, mesenchymal, basal-like immune-suppressed, and basal-like immune-activated[11,12], emphasizing the need for a deeper understanding of the drivers of TNBC to improve treatments for these different patient populations that present with TNBC. Accomplishing these goals requires robust laboratory

models that capture this heterogeneity to support developing new treatment strategies.

Genetically engineered mouse models (GEMMs) that phenocopy breast cancer provide a powerful platform for testing hypotheses regarding tumor development and progression, interaction with the microenvironment, and therapeutic response. Numerous GEMMs have been engineered with inducible, conditional, or constitutively active oncogenes or loss of tumor suppressor genes[13–16]. While useful, these existing models include properties not commonly found in human TNBC, including lack of progression to metastatic disease[17], rare histological features, and non-representative genetic drivers[18]. As such, current TNBC models do not capture the full molecular complexity of human TNBC.

Amplification of the *MYC* oncogene occurs more frequently in TNBC tumors (~60%) than other breast cancer subtypes and is also associated with worse outcomes[19–21]. Additionally, increased phosphorylation of MYC at Ser62 (p-S62-MYC) leads to increased MYC protein stability and transactivation of target genes in human breast tumors and cell lines[22–24]. Upregulation of MAPK and PI3K pathways, frequently observed in breast cancer, leads to increased p-S62-MYC[25]. MYC also stimulates signaling through the PI3K–AKT pathway via upregulation of micro-RNAs that downregulate the tumor suppressor phosphatase *PTEN*[26–29]. In addition, *PTEN* deficiency occurs frequently in TNBC and is linked with aggressive tumors that display high MYC and PI3K pathways and also increased drug resistance[30]. Together, these findings raise the hypothesis that *MYC* gain and *PTEN* loss may cooperate to drive aggressive TNBC.

In this work, we develop a genetically engineered mouse model to replicate *MYC* activation and *PTEN* loss in human TNBC by combining a RosaLSL-Myc/LSL-Myc;Blg-Cre strain with the Ptenfl/fl-conditional knockout mouse model, designated Myc;Ptenfl. The combination of *MYC* deregulation and *PTEN* loss results in the accelerated development of metastatic, heterogeneous triple-negative mammary tumors resembling multiple human TNBC subtypes. We perform comprehensive histological, molecular, and transcriptional analyses together with immune composition and localization to show that Myc;Ptenfl mammary tumors recapitulate inter- and intra-tumoral heterogeneity. Single-cell RNA sequencing (scRNA-seq) reveals differential signatures between the different tumor subtypes, providing insights into putative mechanisms of tumor-microenvironment co-evolution. Together, Myc;Ptenfl tumors effectively recapitulate the different levels of immune microenvironment activity and differential response to standard-of-care therapy observed in human TNBC, highlighting the utility of our Myc;Penfl TNBC model to capture clinically-relevant variation observed in TNBC patients.

## Results

### The combination of deregulated MYC and PTEN loss in mammary epithelium drives rapid triple-negative mammary tumors

Copy number aberrations (CNAs) of *MYC* are frequent in breast cancer. Examining the METABRIC cohort[31] of 2500 primary breast tumors, we found that low-level gain and high-level amplification occur in 48% of all breast cancer (Supplemental Fig. S1A). Among breast cancers classified as TNBC (309/2500)[31], 57% had *MYC* gain or amplification (Fig. 1A and Supplemental Fig. S1A), as well as increased MYC mRNA (Fig. 1B) and decreased survival (Supplemental Fig. S1B, HR = 1.4, 95% CI: 1.0–1.9, $p < 0.05$). Among TNBCs, 36% showed loss of heterozygosity (LOH) or homozygous deletion for *PTEN* (Fig. 1A), resulting in decreased *PTEN* mRNA expression (Fig. 1B). *PTEN* and *MYC* CNAs frequently co-occurred (Fig. 1C, odds ratio (OR) = 4.1, $p < 0.001$), and 65% of TNBCs had altered *MYC*, altered *PTEN*, or both (Fig. 1A). *MYC* CNAs in the presence of *PTEN* loss correlated with poor survival (Fig. 1D).

These observations motivated our investigation of the impact of MYC deregulation and PTEN loss in the mammary gland. We generated Myc;Ptenfl (*Rosa*,*LSL-Myc/LSL-Myc*Pten*,flox/flox*Blg-Cre) mice by crossing the

*Rosa*^LSL-Myc/LSL-Myc^ mice that express two copies of Cre-inducible *Myc* driven by the *Rosa26* promoter, which results in constitutive MYC expression at about twofold above normal mammary gland[32], relevant to the upregulation of MYC in CNA breast cancer (Fig. 1B) with *Pten*^flox/flox^ mice for Cre-inducible knockout of *Pten*[33] (Fig. 1E). The *β*-lactoglobulin-Cre (*Blg-Cre*) transgenic mice were used for mammary-specific Cre expression during late pregnancy and lactation[34]. We monitored tumor development in female mice that had passed two cycles of pregnancy/lactation to induce *Blg-Cre* activation at around 10–12 weeks of age. We compared tumor-free survival of the Myc;Ptenfl mice relative to *Myc* deregulated only and *Pten* loss only mice, all in an FVB genetic background. Mice bearing only the deregulated *Myc* did not develop mammary tumors by 24 weeks after *Blg-Cre* activation (Fig. 1F), consistent with our previous work using *Rosa*,^LSL-Myc/LSL-Myc^*WAP-* or *Blg-Cre* mice where we found homozygous knockin of *Myc* at the *Rosa26* locus was insufficient to drive tumorigenesis after monitoring for 54 weeks[23,32]. *Pten* loss-only mice developed mammary tumors between 110 and 140 days post *Blg-Cre* activation. Combination of deregulated *Myc* with *Pten* loss substantially accelerated tumorigenesis, and these Myc;Ptenfl mice developed mammary tumors between 4 and 135 days, average of 50 days, post *Blg-Cre* activation (Fig. 1F).

We isolated tumors from the *Pten* loss only and Myc;Ptenfl mice and stained them for ER, PR, and HER2. *Pten* loss-only tumors express ERα, PR, and HER2 receptors and show adenosquamous histology, while combination Myc;Ptenfl mammary tumors are 100% triple-negative for these markers (Fig. 1G). In addition, and similar to human TNBC, Myc;Ptenfl tumors show histologic heterogeneity with distinct tumor morphology and degrees of stromal involvement not observed with *Pten* loss alone (Fig. 1G). The Myc;Ptenfl tumors, based on stromal desmoplasia, fall into two broad classes: Stromal-Rich (SR), which has abundant stroma and displays more heterogenous features; and Stromal-Poor (SP), which is a solid-pattern invasive ductal carcinoma (IDC), the most common architectural pattern seen in TNBC patients[35] (Fig. 1G). Staining by immunohistochemistry (IHC) also revealed that SR tumors express higher stromal collagen by Trichrome stain and smooth muscle actin (SMA), basal marker Cytokeratin 5 (KRT5) and phosphorylated Smad3, which can promote epithelial–mesenchymal transition (EMT)[36], compared with SP tumors (Supplemental Fig. S1C). We also observed increased expression of post-translationally stabilized S62 phosphorylated MYC in SP relative to SR tumors (Supplemental Fig. S1D).

Along with accelerated tumor onset and triple-negative status, Myc;Ptenfl tumors were also more metastatic at the IACUC-defined endpoint, with a 52% and 60% metastasis rate to the lymph node and/or lung, in SR and SP, respectively, compared with only 16% overall metastatic rate for Ptenfl endpoint tumors (Fig. 1H, and Supplement Fig. S1E). Together, these data indicate that the Myc;Ptenfl mice are an in vivo model of heterogeneous, aggressive TNBC.

### Myc;Ptenfl tumors show molecular and histologic subtype heterogeneity

We performed gene expression profiling on 13 Myc;Ptenfl tumors and 3 control (no Cre) normal mammary glands by RNA sequencing (RNA-Seq) to examine their molecular characteristics. Unsupervised hierarchical clustering on normalized gene expression data recapitulated the SP and SR histologic groupings (Fig. 2A, clusters 1 vs. 2), with several molecular subclusters for SR (cluster2), which is the more heterogenous subtype. Genotype-blinded analysis by two board-certified pathologists identified histologically distinct features in the SR tumors that molecularly subclustered within the SR cluster (Fig. 2A). These included IDC with lobular features, sufficient squamous differentiation to be classified as IDC with squamous features, and metaplastic IDC with mixed cell phenotypes including spindle-like cells (Fig. 2B, C). SP tumors were marked as solid IDC with little stromal desmoplasia (Fig. 2B, C). In a histological analysis of 123 Myc;Ptenfl

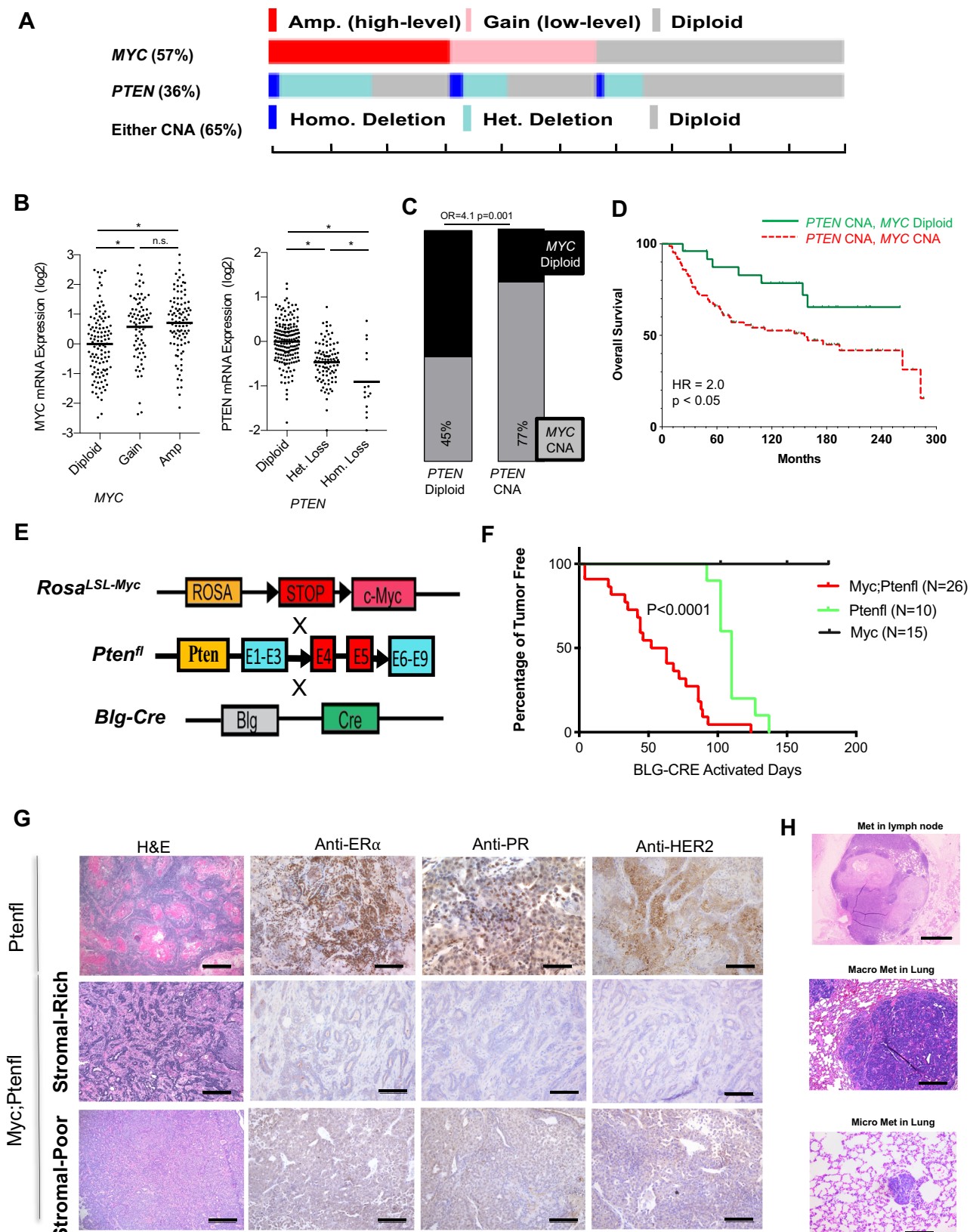

tumors, we found a distribution of 77% SR subtypes (predominately IDC with lobular features 60%, squamous 15% and metaplastic 2%) and 23% SP subtype tumors (Fig. 2C). Thus, they follow human tumors where metaplastic is rare and accounts for 0.2–5% of all breast cancers[37]. The two distinct histologic subtypes are also separated via principal component analysis (Fig. 2D, E). Overall, the bulk

transcriptional analysis suggests that the SP samples are associated with a tightly regulated transcriptional state, whereas the stromal-rich samples demonstrate more heterogeneity in both principal component and gene space.

Mice-bearing SR tumors developed tumors earlier than SP, about 52 days after *Blg-Cre* activation (Fig. 2F), however, these SR tumors

**Fig. 1 | Deregulated *Myc* combination with delated *Pten* in mammary gland accelerates triple-negative mammary tumorigenesis. A** Copy number alteration (CNA) in *MYC* showing amplification or gain and *PTEN* showing shallow or deep deletion and **B** mRNA expression of MYC and PTEN in 309 ER−/HER2− patients from 2500 breast cancer patients–METABRIC Data[31] (* represent significant two-tailed *p* value < 0.05, n.s represent a non-significant *p* value). **C** Total 101 *PTEN* deletion in 309 ER-/HER2- patients with 77% MYC amplification or gain; the rest, 198 ER-/HER2- *PTEN* diploid patients, with 45% *MYC* amplification or gain, (two-tailed *p* value = 0.001). **D** Survival in 101 ER−/HER2− patients from METABRIC Data with *PTEN* loss and with (red dotted line) or without (green line) *MYC* amplification or gain (*p* value = 0.0187 using Gehan−Breslow−Wilcoxon test). **E** Diagram for generation of Myc;Ptenfl (Rosa;$^{LSL-Myc/LSL-Myc}$Pten;$^{fl/fl}$Blg-Cre) mice by breeding the Rosa$^{LSL-Myc/LSL-Myc}$ conditional knockin and Pten$^{fl/fl}$ conditional knock-out mice with Blg-Cre transgenic mice. **F** Mammary gland tumor incidence from Myc (Rosa;$^{LSL-Myc/LSL-Myc}$Blg-Cre), Ptenfl (Pten;$^{fl/fl}$Blg-Cre) and Myc;Ptenfl mice post-breeding and lactation for Blg-Cre activation (*p* value = 0.00001 using Gehan−Breslow−Wilcoxon test). Myc;Ptenfl (*N* = 26), Ptenfl (*N* = 10), and Myc (*N* = 15). **G** H&E staining for Myc;Ptenfl tumors, and Immunohistochemistry staining with anti-PR, anti-HER2, and Anti-estrogen receptor a (ERa). Representative images of 27 mammary gland tumors from Myc;Ptenfl mice and 10 tumors from Ptenfl mice. Scale bar = 100 μm. **H** Representative H&E staining for macro lymph node and lung metastases and micro lung metastasis from 38 Myc;Ptenfl mice and 19 Ptenfl mice. Metastasis rates: Ptenfl: 3/19 = 16%. Myc;Ptenfl stromal-rich: Macro 9/23 = 39%; Micro 3/23 = 13%. Myc;Ptenfl stromal-poor: Macro 5/15 = 33.3%; Micro 4/15 = 26.7%. Scale bars = 1 mm (Top), 200 μm (middle), 100 μm (bottom).

took an average of 55 days to reach 2 cm diameter tumors (Fig. 2G). Although mice bearing SP tumors took an average of 102 days after *Blg-Cre* activation to develop tumors (Fig. 2F), SP tumors grew faster than SR tumors, with an average of 33 days to reach the 2 cm diameter end-point size (Fig. 2G). We also examined the response to the commonly used standard-o-care chemotherapy, Paclitaxel (Fig. 2H). Mice bearing SR tumors were sensitive to Paclitaxel, while mice with SP tumors displayed resistance; histology was determined from end-point tumors (Fig. 2H). The ratios of SP and SR tumors found after therapy are comparable to the ratios in the control-treated mice and in the model overall. As a result, even though the histology of these tumors prior to treatment remains unknown, we anticipate that it has not changed. Altogether, SP tumors are the more aggressive subtype with poor OS and resistance to standard-of-care therapy as compared to SR tumors.

To investigate whether the histologically heterogeneous Myc;Ptenfl subtypes originated from similar or distinct tumor phenotypes at an early stage, we examined 32 small (diameter < 3 mm) tumors. We observed solid ductal carcinoma and SR histologies, including papillary, lobular, and adenosquamous, with high expression of the basal marker KRT14; and all were ER, PR, and HER2 negative (Supplemental Fig. S2A–C). These data suggest that this model may provide a unique resource for examining early events that generate distinct TNBC subtypes.

We used genes set enrichment analysis (GSEA)[38] to assess differences in Cancer Hallmark[39] pathway activity between Myc;Ptenfl SR and SP tumors. This demonstrated that the SR tumors are enriched for pathways related to EMT, angiogenesis, allograft rejection, and inflammatory response, whereas the SP tumors are enriched in gene sets related to Interferon (IFN) response and oxidative phosphorylation (OXPHOS) (Fig. 2I and Supplemental Fig. S3A, Source Data). OXPHOS is important for the production of biosynthetic intermediates necessary to support the rapid proliferation of cancer cells and associated with high lethality in TNBC[40], which may explain the high growth rate and poor prognoses observed in the SP subtype.

We assessed the expression of published gene signatures that classify human TNBCs into one of four subtypes[12]. The correlation between our Myc;Ptenfl tumor samples and the gene signature centroids demonstrated that the SR tumors were more correlated with the human TNBC Mesenchymal (MES) subtype, whereas SP tumors were better correlated with the basal like immune activated (BLIA) and basal like immune suppressed (BLIS) human TNBC subtypes (Fig. 2J). The SR assigned MycPten;fl tumor (sample b11) was both an outlier for correlation to the LAR subtype (spearman correlation = 0.3, mean across all samples = −0.02) and PC2 embedding, suggesting the pattern picked up by PC2 may be related to the LAR TNBC subtype. IHC staining for AR agreed with the lack of correlation to the human luminal androgen receptor (LAR) subtype, with <1% AR staining in both SP and SR tumors (Supplemental Fig. S3B).

GSEA results (Fig. 2I) indicated that inflammatory response is one of the top Hallmark pathways enriched in the SR phenotype, which has a better survival post tumor detection. SP was enriched with IFN response genes, which have been associated with metastatic TNBC[41]. To investigate if immune cells might be differentially recruited across Myc;Ptenfl tumor subtypes, we evaluated the expression of chemotactic cytokines among the subgroups of Myc;Ptenfl mouse model tumors[42]. The SR subtype had significantly higher (Log2FC > 1.5 and padj < 0.05) chemotactic chemokine gene expression than the SP subgroup for 5 out of 7 chemotactic cytokines associated with prognosis in human breast cancer (Fig. 2K).

## Multiplexed immunohistochemistry platform identifies distinct immune complexity profiles across Myc;Ptefl subtype tumors

We utilize a multiplexed immunohistochemistry (mIHC) approach to characterize the immune contexture and the spatial distribution of immune cells among the subgroups of the Myc;Ptenfl mouse tumors. The mIHC platform comprises a validated panel of 23 antibodies in a sequential staining method for the identification of lymphoid and myeloid immune cell lineages, functional markers, and epithelial markers in a single FFPE tissue section (Fig. 3A and Source Data)[43,44]. Qualitatively, SR tumors show higher infiltration of lymphoid and myeloid immune cell lineage, including T-cells markers–(CD3, CD8, and CD4), B cells marker (CD45R), Treg cell marker (Foxp3), and macrophages marker (CD68) than SP tumors (Fig. 3A, B).

We utilized an image analysis pipeline[45] for quantitative assessment of the 23-plex images and analyzed three spatial regions: tumor periphery, tumor border, and tumor core[46] (Fig. 3B, Supplemental Fig. S4). As predicted, for both subgroups, we observed the lowest density of CD45+ immune cells in the tumor core as compared to the border and periphery regions (Fig. 3B). However, SR tumors had significantly higher total immune cell density in all tumor cores, border, and periphery compartments compared to SP. We employed an unsupervised hierarchical clustering approach and observed two distinct immune complexity profiles for SR and SP tumors across spatial compartments, where lymphoid and myeloid lineage cells were differentially present between the groups, indicating distinct immune contexture by subtype (Fig. 3B). When performing supervised analyses, we observed that SR tumors had trending higher densities of Ly6G$^+$ granulocytes, B220$^+$ B cells, CD11c$^+$ DCs, and CD4$^+$ and CD8$^+$ T cells in all spatial compartments, with significantly higher density of Ly6G$^+$ granulocytes and CD8$^+$ T and DC cells in tumor border and periphery, respectively (Supplemental Fig. S4C–E). The proportion of FoxP3$^+$ Tregs within the CD4$^+$ T cells was greater in SR (Supplemental Fig. S4F, G). Sankey diagrams showing the relative density of the indicated cell types by spatial compartment indicate that PanCK$^+$ neoplastic cells dominate in all spatial categories with an increase toward the tumor core in both subtypes (Fig. 3C). As expected, the SP group shows a very high abundance of PanCK$^+$ neoplastic cells in the tumor core. Notably, CD4$^+$ and CD8$^+$ T cells (light blue and dark blue lines, respectively) are most abundant in the periphery and border, particularly in SR tumors, dropping in

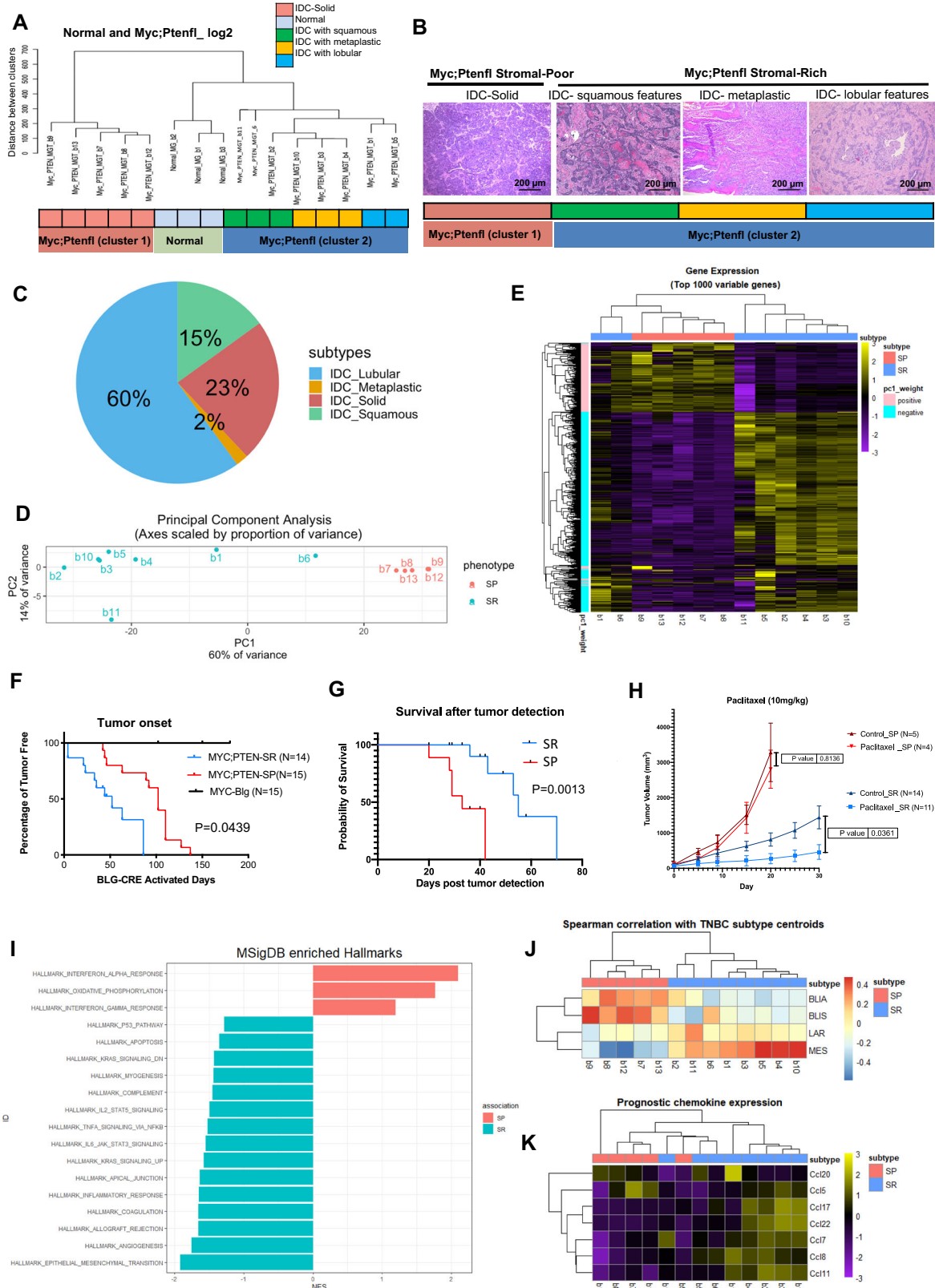

abundance in the tumor core, while macrophages dominate in all spatial categories (maroon line) (Fig. 3C). SR tumors contain a higher abundance of multiple immune cell types and trends toward more lymphocytes compared to SP tumors. Particularly, CD8+ T cells are significantly higher in the SR tumor periphery than in the SP periphery (Supplemental Fig. 4E, *p*-value = 0.05). Thus, SR tumor immune contexture is consistent with studies showing that high levels of stromal T cells are associated with improved OS and better response to therapy in human TNBC[11,47,48]. These findings indicate that the Myc;Ptenfl model may provide a unique tool for assessing a spectrum of immune cell mechanisms in triple-negative tumor biology and drug response.

**Fig. 2 | Multiple molecular and histologic subtypes are present in Myc;Ptenfl tumors with human TNBC subtype-specific transcriptomic signatures.** **A** Unsupervised hierarchical clustering of RNA expression from 13 Myc;Ptenfl mammary gland tumors and 3 normal mammary glands. Tumor clusters and histologies indicated. **B** Histology of tumors from **A** in cluster 1-solid (reproduce and represent 28 mice bearing tumor) and cluster 2 (reproduce and represent a total of 95 mice bearing tumor; 18 squamous, 3 Metaplastic, and 74 lobular features). Scale bars = 200 μm. **C** Pie chart of an extended histological analysis of 123 tumors from Myc;Ptenfl mice indicating the frequency of histological subtypes. **D** Principal Component Analysis of RNAseq from 13 Myc;Ptenfl tumors; SR (blue), SP (red). Axes scaled by the proportion of variance (PC value divided by the proportion of variance for that principal component). **E** Gene expression heatmap for top 1000 variable genes (genes used for PCA), color-coded to indicate positive or negative weight for PC1. Gene expression is computed as counts with VST normalization and then z-scored across samples. **F** Days from the end of pregnancy/lactation, when Blg-Cre activated, to detection of mammary gland tumor in Myc;Ptenfl mice. Tumor histology group stromal-rich and stromal-poor indicated. Stromal-Poor (red, N = 15), Stromal-Rich (blue, N = 14), and Myc-Blg (black, N = 15).

**G** Survival after tumor detection for stromal-rich tumor-bearing mice vs. the mice with stromal-poor tumors. Stromal-Poor (SP) (red, N = 6) and Stromal-Rich (SR) (blue, N = 5). **H** Tumor volume during Paclitaxel treatment (10 mg/kg); Stromal-Poor (red, N = 4) and Stromal-Rich (blue, N = 11), or vehicle-treated controls; Stromal-Poor (dark red, N = 5) and Stromal-Rich (dark blue, N = 14), Data are presented as mean values ± SD. **F**−**H** using Gehan−Breslow−Wilcoxon test. **I** Bar plot to visualize the significantly enriched (Adjusted p value < 0.05) MSigDB hallmarks for SR vs. SP differentially expressed genes. The x-axis is the normalized enrichment score, and y-axis shows the enriched hallmark geneset (Source data are provided as a Source Data file). **J** Spearman correlation of MycPten;fl tumors to human TNBC subtype centroids. Centroid signatures filtered to 60 homologous genes mapped between mouse and human transcriptomes (60/77 = 78%) to identify the similarity of MycPten;fl subtypes to four prognostically-distinct human TNBC subtypes; basal-like immune-activated (BLIA), basal-like immunosuppressed (BLIS), luminal androgen receptor (LAR), and mesenchymal (MES). **K** Gene expression heatmap for seven chemotactic cytokines associated with prognosis in human breast cancer. Gene expression is computed as counts with VST normalization and then z-scored across samples.

## Morphological feature extraction identifies shared morphologies between human breast cancer and Myc;Ptenfl mammary tumors

To compare the morphological features of Myc;Ptenfl tumors with human patient TNBC, we compared several tissue microarrays (TMAs), one generated from our TNBC mouse model and two generated from human breast cancer. To assess the morphological features, TMA H&E images were tiled into 28 regions per core. We used a variational autoencoder (VAE)[49,50]−an unsupervised deep learning (DL)-based method for representation learning and feature extraction (Fig. 4A, Supplemental Fig. S5A). From each normalized H&E tile, we extracted a morphological feature vector to establish a comparison between each tile, cores, and tissue origins. Tiles were compared using UMAP embedding[51,52] as well as k-means clustering analysis of the latent feature vectors. The regional H&E tile images on UMAP showed distinct morphological feature differences, and density maps for human and mouse tiles showed overlapping regions in UMAP embedding space, highlighting shared morphological features between human and mouse TNBC (Fig. 4B). K-means clustering identified eight representative groups (Fig. 4C). Representative tile images near the cluster center and single high-resolution tiles within each cluster illustrate their distinct morphologies, including carcinoma with discohesive growth pattern (cluster a), IDC high grade (cluster d), and fibrotic stroma (cluster f) (Fig. 4D). Overall, quantitative cluster analysis shows a high level of overlap between human and mouse tiles (Fig. 4E). The most prominent human morphologies (clusters a, d, and f), which comprise 72% of all human tiles, also have a high representation of mouse tiles. Clusters that show a high class imbalance (clusters b and e) comprise a much smaller portion of the total tile population. While TNBC was the predominant subtype on the human TMAs, ER+ and TNBC subtypes overlapped with each other in morphologies identified by the VAE (Supplemental Fig. S5B).

We compared histologic features of mice and human TMAs annotated by a pathologist (Supplemental Fig. S6 and Source Data). The Myc;Ptenfl tumors show a diverse histologic spectrum from ductal carcinoma not otherwise specified (NOS) to various metaplastic elements, similar to the typical characteristics of human TNBC[53,54]. Myc:Ptenfl tumors showed IDC, histologic grade 2 (low-grade nuclear features) (21.25%), histologic grade 3 (high-grade nuclear features) with marked nuclear pleomorphism and prominent nucleoli (17.5%), solid sheet-like growth pattern without tubule formation (12.5%), stromal proliferation with lymphocytic infiltrate (33.75%), geographic necrosis (21.25%)[55], and myoepithelial proliferation (41.25%). Like human TNBC, various metaplastic changes such as sarcomatoid change with spindle cells (12.5%) and squamoid change with keratin pearls (47.5%) were observed[56]. Although it was not identified in the

human TNBC TMA cores of this study, clear cell change (11.25%)[57] and thick trabeculae suggestive of neuroendocrine differentiation (1.25%)[58], which are reported as very rare human TNBC, were also observed in the Myc;Ptenfl samples.

Notably, the most abundant histological features in the Myc;Ptenfl SR subtype are squamoid metaplasia, myoepithelial proliferation, and stromal lymphocytic infiltrate with 59.37%, 51.56%, and 39% frequency, respectively. Although metaplastic tumors have been shown to be highly chemoresistant and aggressive, the stromal lymphocytic infiltrate phenotype has been associated with improved survival[59–63]. In contrast, the most frequent histological features in the Myc;Ptenfl SP phenotype, which has a poorer prognosis, are low- and high-grade nuclear features and a solid growth pattern with 50%, 37.5%, and 50% frequency, respectively. Indeed, human TNBC studies showed that a solid growth pattern and high-grade nuclear features are typically associated with poor TNBC prognosis, resistance to standard-of-care therapy, and histological features of aggressive TNBC[59–62]. These results indicate that our mouse model recapitulates the histological heterogeneity and corresponding spectrum of prognostic features seen in TNBC patients.

## Shared tumor and microenvironment cellular phenotypes in Myc;Ptenfl TNBC tumors and human TNBC

We used cyclic immunofluorescence (CyCIF) with 20 markers to examine epithelial and stromal cell phenotypes in Myc;Ptenfl TNBC tumors (Source Data). CyCIF staining of the Myc;Ptenfl TMA confirmed the histologic SR and SP subtypes with increased stromal cells in the SR tumors and increased epithelial cells in the SP tumors (Fig. 5A). Quantitative image analysis with the mplexable software[64], followed by gating on cell-type specific markers determined frequencies of epithelial, immune, and stromal cells (Fig. 5B and Supplemental Fig. S7A−C). To compare our data to human TNBC, we obtained a publicly available human TNBC multiplex ion beam imaging (MIBI) dataset[65]. Gating on cell-type specific markers identified epithelial, immune, and stromal cells in the human tissue (Supplemental Fig. S7D−H). Clustering of mice and human tissues based on cell frequency and further annotation by the levels of epithelial, immune, and stromal non-immune cell types into clusters revealed three general phenotypes: SP, SR-immune-rich (SR-IR), and SR-immune-poor (SR-IP) (see Methods, Fig. 5C−E, and Supplemental Fig. S8A−G). Further splitting the SP phenotype into SP+ with the lowest stroma demonstrated significant association with the mouse histology SP and SR subgroup designations (Supplemental Fig. S8D, E) (chi-squared p = 2.2e−6). Importantly, the mean frequency of epithelial, immune, and stromal cell types in each of the three SR-IR, SR-IP, and SP subtypes was similar between mouse and human TNBC (Fig. 5E). Additionally,

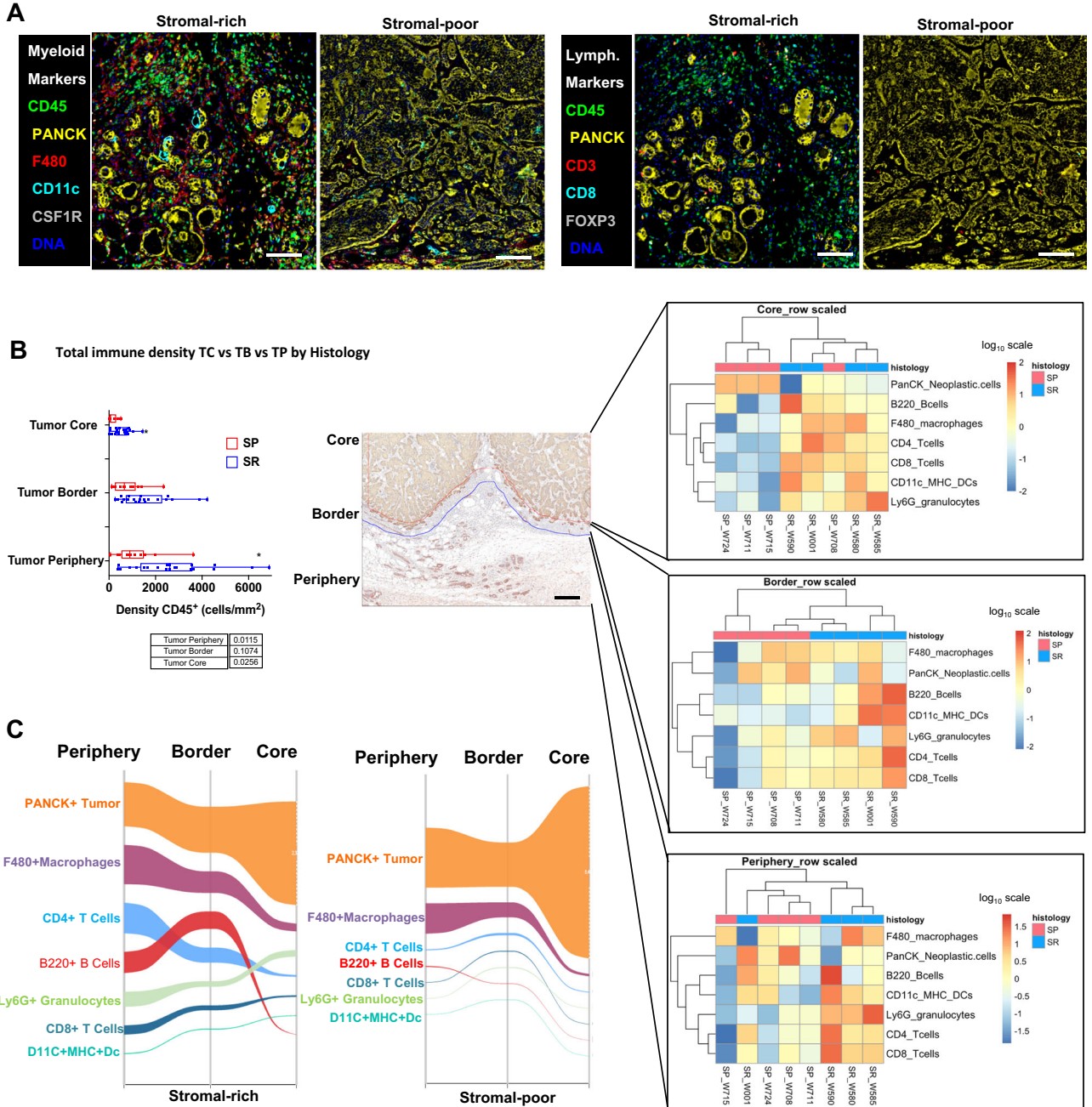

**Fig. 3 | Immune contexture across Myc;Ptenfl subtypes. A** Example of multiplex immunohistochemistry (mIHC) images from Myc;Ptenfl Stromal-rich (*n* = 21) and Stromal-poor tumors (*n* = 13) with Myeloid markers expression on the expression of the left and lymphoid marker on the right. Scale bars = 100 μm. **B** Total immune cells by CD45⁺ density in Stromal-rich tumor (*n* = 21 biologically independent samples) periphery, border, and core compared to Stromal-poor tumors (*n* = 13 biologically independent samples). Left graph; box-and-whisker plots show median and interquartile range (* represent significant *p* value < 0.05 using two-way ANOVA (mixed model)), the centerline of the boxplots represents the median value (50th percentile), and the box encapsulates the range from the 25th to the 75th percentiles of the dataset. The whiskers extend from the minimum to the maximum values, showcasing the full spread of the data. Middle image; Tumor border was determined by CD45 and PANCK expression (supplemental S4A and B. Scale bars = 100,000 μm). On the right, unsupervised clustering; heatmap color computed as the z-score of log10 normalized celltype density (cells/mm²) in each periphery, border, and core of Myc;Ptenfl tumor subtypes. **C** Sankey diagrams showing the distribution of each immune lineage population in stromal-rich (left) and stromal-poor (right) tumors; periphery, border, and core.

the SP and SR subtypes were prognostic in human MIBI data, with SP having shorter OS compared with the SR subtypes, consistent with subtype survival in our TNBC mouse model (Fig. 5F, Supplemental Fig. S8H).

We compared the mean single-cell expression of markers in stromal and epithelial cells in the three subtypes in human and mouse tumors and found similar differences (Fig. 5G–J, Supplemental Fig. S9A, B). In both human and mouse tumors, the SR-IR subtype had

greater stromal pan-immune CD45, T regulatory FOXP3, and dendritic CD11c expression, consistent with the multiplex IHC platform (Fig. 5G, I). In mouse and human SP tumors, the stromal cells expressed relatively more of the endothelial marker CD31 (Fig. 5G, I). Stromal proliferation, as measured by Ki67 expression, was lowest in SR-IP tumors in mice and humans, but there was no difference in epithelial Ki67 (Fig. 5G–J). Human and mouse epithelial cells from SR-IR tumors showed higher expression of the mesenchymal marker vimentin

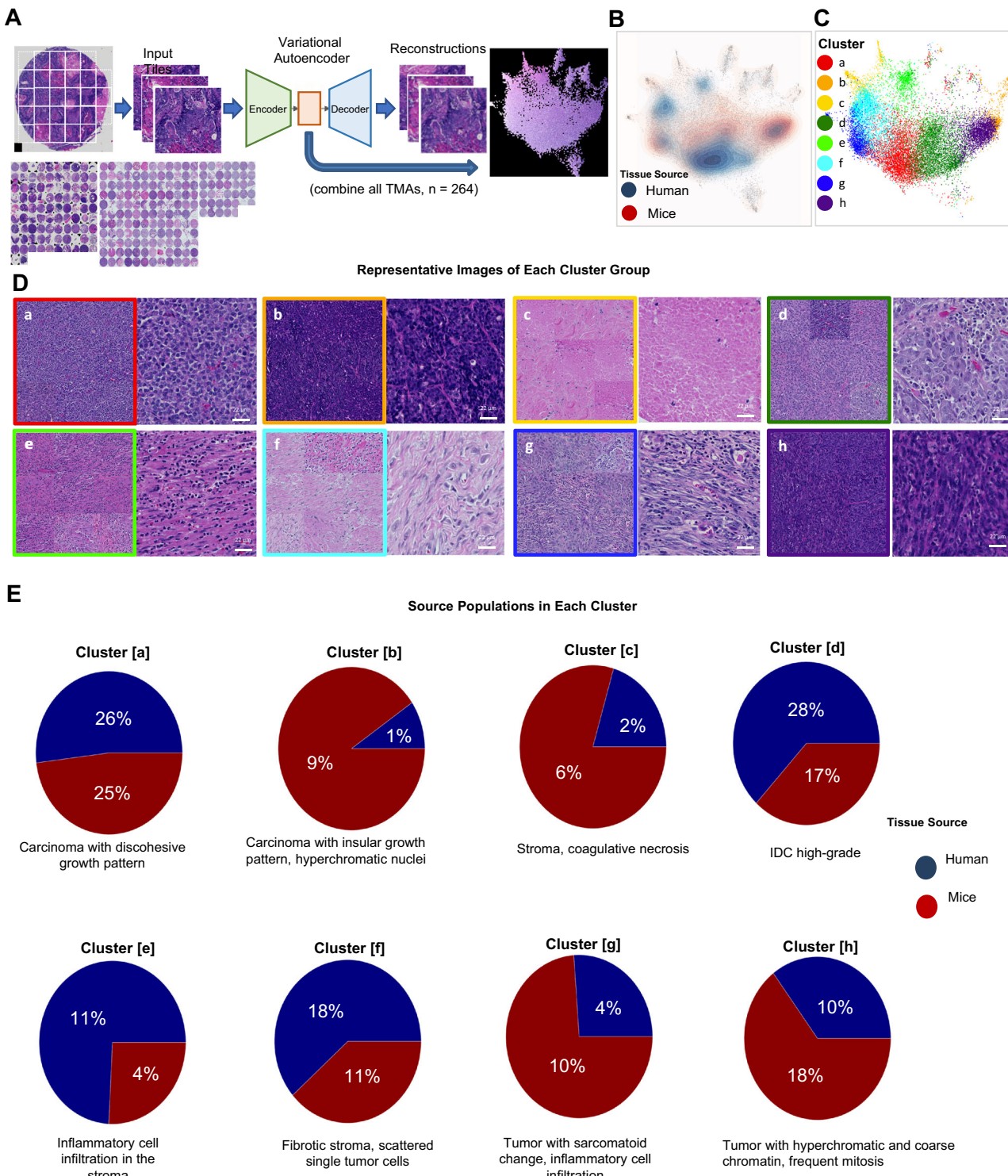

Fig. 4 | Analysis of shared morphologies between human breast cancer and Myc;Ptenfl tumor TMAs. A Pipeline for generating morphological feature representation using a variational autoencoder (VAE). Tiles from both mice and human TMAs are used to train a VAE, then a latent encoding vector is computed for each tile. Tiles are compared using UMAP embedding and k-means clustering analysis of the latent features. B Density functions for all human and mouse tumor tiles are calculated within the two-dimensional UMAP space to visually compare overlap in embedding space (corresponding histopathological feature image is shown in Supplemental Fig. S5A). C K-means clusters (*n* = 8) are computed using latent features and projected into UMAP space for visualization. Clusters consisting primarily of edge artifacts were excluded from the analysis. D For every cluster, the nine tiles closest to the cluster center (left) and a single high-resolution tile image within each cluster (right, a representation of the nine tiles clusters. Scale bar = 22 μm) are shown to illustrate each cluster dominant morphology; main histologic features of each center: [a] carcinoma with discohesive growth pattern; [b] carcinoma with thin fibrotic septa and hyperchromatic nuclei; [c] stroma or coagulative necrosis; [d] IDC with high-grade nuclear feature; [e] inflammatory cell infiltration in the stroma; [f] fibrotic stroma, scattered single tumor cells; [g] sarcomatoid change of tumor cells and inflammatory cell infiltration; and [h] tumor with hyperchromatic and coarse chromatin with frequent atypical mitosis (The figure includes 92 mice TMA cores, representative of 80 mice, and 172 human TMA cores, corresponding to 172 patients). E Relative abundance of human and mouse tumor tiles is calculated for each cluster using the ratio of tiles in a cluster to total tiles from the given TMA source.

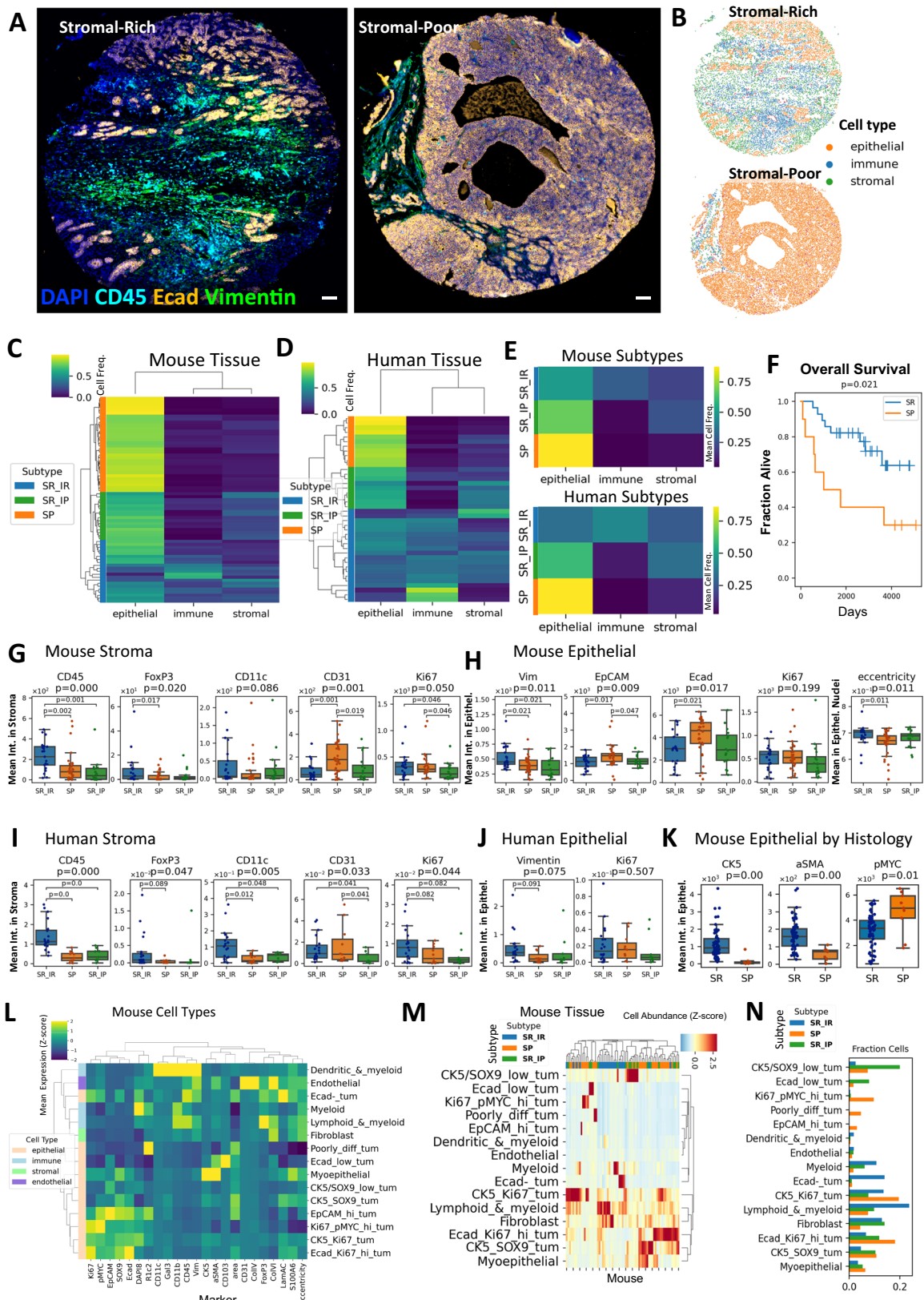

(Fig. 5H, J). Mouse SR subtypes had higher nuclear eccentricity and lower expression of the epithelial markers E-cadherin and EpCAM compared to SP (Fig. 5H), consistent with the mouse bulk RNAseq data demonstrating that SR tumors had more mesenchymal gene expression and correlated with the mesenchymal human TNBC subtype. Consistent with the original IHC staining (Supplemental Fig. S1C, D),

the histologically designated SR subtype showed high expression of αSMA and the basal marker CK5 compared to the SP, while phospho-MYC was higher in the SP subtype epithelial cells (Fig. 5K).

We further analyzed the co-expression of phenotypic markers in the mouse TMA via unsupervised clustering of single cells with the Leiden algorithm (Supplementary Fig. S7C). This resulted in 21 clusters

**Fig. 5 | Tumor and microenvironment cell phenotype comparisons between Myc;Ptenfl tumors and human TNBC. A** Cyclic immunofluorescence (CyCIF) staining of representative Myc;Ptenfl tissue microarray (TMA) cores (1.5 mm diameter) of stromal-rich (left, reproduced n = 59) and stromal-poor (right, reproduced n = 10) histology subtypes with the indicated markers. Scale bar = 130 μm. **B** Cell type calling defined by gating on cores in (**A**). **C** Hierarchical clustering of Myc;Ptenfl tumor samples based on cell type frequency in each tumor core. **D** Hierarchical clustering of human TNBC samples based on cell type frequency in each region of interest in TNBC tumors imaged with multiplex ion-beam imaging (MIBI). **E** Mean frequency of cell types in cell frequency-based subtypes in Myc;Ptenfl (top) and human TNBC (bottom) samples. **C–E**. Heat map row colors: stromal-poor (SP, orange), stromal-rich-immune-rich (SR_IR, blue), and stromal-rich-immune-poor (SR_IP, green) subtypes. **F** Kaplan–Meier curves of overall survival in cell frequency-based subtypes in human TNBC MIBI data. Log-rank p = 0.021, n = 38 patients, vertical ticks are censored patients. **G** Stromal expression of pan-immune (CD45), Treg (FoxP3), dendritic (CD11c), endothelial (CD31), and proliferation (Ki67) markers in mouse 3-class subtypes. **H** Epithelial expression of mesenchymal (Vim), epithelial (EpCAM, Ecad), proliferation, and nuclear eccentricity markers in mouse subtypes. **I** Stromal expression of pan-immune, Treg, dendritic, endothelial, and proliferation markers in human subtypes. **J** Epithelial expression of mesenchymal (Vimentin) and proliferation markers in human subtypes. **G–J** P-values are determined by the Kruskal–Wallis H test; the centerline of the boxplots represents the median value (50th percentile), and the box encapsulates the interquartile range. The whiskers extend to show the rest of the distribution, except for outliers defined as 1.5 times the interquartile range. Dots overlaid on boxplots show individual cores' mean; N = 70 mouse TMA cores (**G, H, K**) and 40 human patients (**I, J**). **K** Epithelial expression of basal/myoepithelial markers (CK5, alpha-SMA) and phospho-MYC in mouse histological subtypes. P-values determined by Mann–Whitney U test. the centerline of the boxplots represents the median value (50th percentile), and the box encapsulates the interquartile range. The whiskers extend to show the rest of the distribution, except for outliers defined as 1.5 times the interquartile range. Stromal-rich (n = 59) and stromal-poor (n = 10). **L** Mean marker intensity in each annotated cell type defined by unsupervised Leiden clustering in Myc;Ptenfl tissues. **M** Hierarchical clustering of mouse samples based on detailed cell types. Heat map column colors: stromal-poor (SP, orange), stromal-rich-immune-rich (SR_IR, blue), and stromal-rich-immune-poor (SR_IP, green). **N** Bar plot of the frequency of each cell type in mouse subtypes.

which, after artifact removal and combining of similar clusters, produced 14 distinct annotated cell type clusters (Fig. 5L, Supplemental Fig. S7C). These cell types captured intratumoral heterogeneity observed in the images (Supplemental Fig. S10A-B). Hierarchical clustering of tissues based on z-scored cell abundance revealed phenotypic heterogeneity within the SP and SR-IP subtypes (Fig. 5M). Conversely, most of the SR-IR tissues clustered together, driven by a high abundance of fibroblasts and immune cells in the tissues (Fig. 5M). Consistent with our other analyses, SR-IR tumors were enriched in immune cells, had fewer Ki67+ proliferating tumor cells, and contained E-cadherin negative epithelial cells, fitting with a mesenchymal phenotype (Fig. 5N). SP tumors were enriched in Ki67+ tumor cells and de-enriched in fibroblasts and E-cadherin low/negative tumors, consistent with their fast-growing, solid pattern (Fig. 5N). Single cells from the human MIBI data were clustered with the Leiden algorithm resulting in 25 cell-type clusters that were annotated as epithelial, immune, or stromal (Supplemental Fig. S7F). Similar to mice, human SR-IR tumors were enriched in immune cell type clusters, with similar patterns of heterogeneity (Supplemental Fig. S10C–E).

### scRNA-seq reveals Myc;Ptenfl tumor subtype-specific distributions of cell states

We performed deep single-cell RNA sequencing (scRNA-seq) of 11 MycPten;fl tumors (4 SP, 7 SR) (Supplemental Fig. S11A–C. After quality control thresholding, we retained a total of 14,042 cells (490–5995 cells per tumor) (Supplemental Fig. S12A–C). We used integrative non-negative matrix factorization (INMF) to identify patterns of transcriptional expression describing malignant and stromal cells within MycPten;fl tumors (Supplemental Fig. S13A, B)[66]. We identified cell clusters using the Leiden algorithm applied to iNMF embeddings with parameters optimized for maximum silhouette width[67,68]. We manually assigned cell lineages (epithelial, lymphoid, myeloid, fibroblast, endothelial, and perivascular) based on the aggregated expression of canonical markers (Fig. 6A, Source Data). One cluster (cluster 14) was defined by expression of cell cycle-related genes rather than lineage signatures, and this cluster was further subclustered to assign each subcluster to an appropriate lineage (Supplemental Fig. S13C–E). Consistent with CyCIF, SR tumors formed two groups: one with a high representation of both immune and non-immune stromal cell types (SR-IR) and another with a high representation of non-immune stromal cells but low immune cell representation (SR-IP) (Fig. 6B and see Supplemental Fig. S13F, G for UMAP and cell frequency representations by subtype).

To investigate lineage heterogeneity, we performed differential expression analysis between clusters in same-lineage clusters. Putative cell types were assigned using an automated cell type classifier and then manually refined using canonical biomarker genes (Fig. 6C, D, Supplemental Fig. S14A–C, Source Data)[69,70]. Enrichment of Gene Ontology analysis was performed on the differentially upregulated genes for each cluster (average log2 fold change > 0.5 and Bonferroni adjusted p ≤ 0.05), and these clusters were assigned names to reflect the themes of their most enriched ontologies (Fig. 6C, D). We observed clear distinctions in epithelial and fibroblast cell-type proportions across the three tumor subtypes, suggesting that the different epithelial and fibroblast states may be involved in stromal expansion and immune exclusion or recruitment (Fig. 6C, D, Supplemental Fig. S14D). Epithelial cluster 2 was the dominant epithelial state for SP tumors (71% of SP epithelial cells) and was enriched for metabolism-related programs, including upregulation of OXPHOS-related ontologies and luminal markers (Fig. 6E, F, Supplemental Fig. S14D, E), which provides further evidence for SP tumors recapitulating OXPHOS high human TNBC subtypes. SR-IP tumors instead were mostly comprised of epithelial cells belonging to either cluster 8 or 10 (51% and 44% of SR-IP epithelial cells, respectively, Fig. 6C, D, Supplemental Fig. S14D). Epithelial cluster 8 was enriched for ontologies related to gland development and had high expression of several luminal genes (Supplemental Fig. S14E)[71]. Epithelial cluster 10 had enriched ontologies associated with extracellular matrix modulation and high expression of Krt5 and Krt14, both of which are associated with a basal-like TNBC subtype (Fig. 6E, F, Supplemental Fig. S14E)[71]. The final epithelial cluster 19 was found at the highest rate in SR-IR tumors (27% of SR-IR epithelial cells, <1% of SR-IP epithelial cells, 2% of SP epithelial cells) and had enriched ontologies for reactive oxygen species response and regulation of apoptotic signaling pathways (Fig. 6C–F, Supplemental Fig. S14D).

The Myc;Ptenfl tumors showed a similar subtype-specific enrichment of different fibroblast clusters. SP tumors were enriched for cluster 7 fibroblasts, which had high Ctla2a expression (Fig. 6C, G, H). High Ctla2a in fibroblast has been shown to be associated with immune suppression by induction of apoptosis for T-cell lymphocytes[72]. SR-IP tumors were found to have a higher fraction of cluster 12 fibroblasts which uniquely expressed the CAF marker Col11 and had upregulated gene activity related to ossification and apoptosis pathway regulation (Fig. 6C, G, H)[73]. Fibroblast cluster 17 included signaling related to ameboid cell migration and coagulation and was more common in both the SR-IP and SR-IR subtypes (Fig. 6C, G, H).

Consistent with mIHC, macrophage cells were the most abundant immune cells across Myc;Ptenfl subtypes. The SR-IR subtype had a higher fraction of cells coming from both lymphoid and myeloid lineages, including neutrophils (cluster 3, Lrg-high and cluster 4, Ccl3-high), when compared to SR-IP and SP tumors.

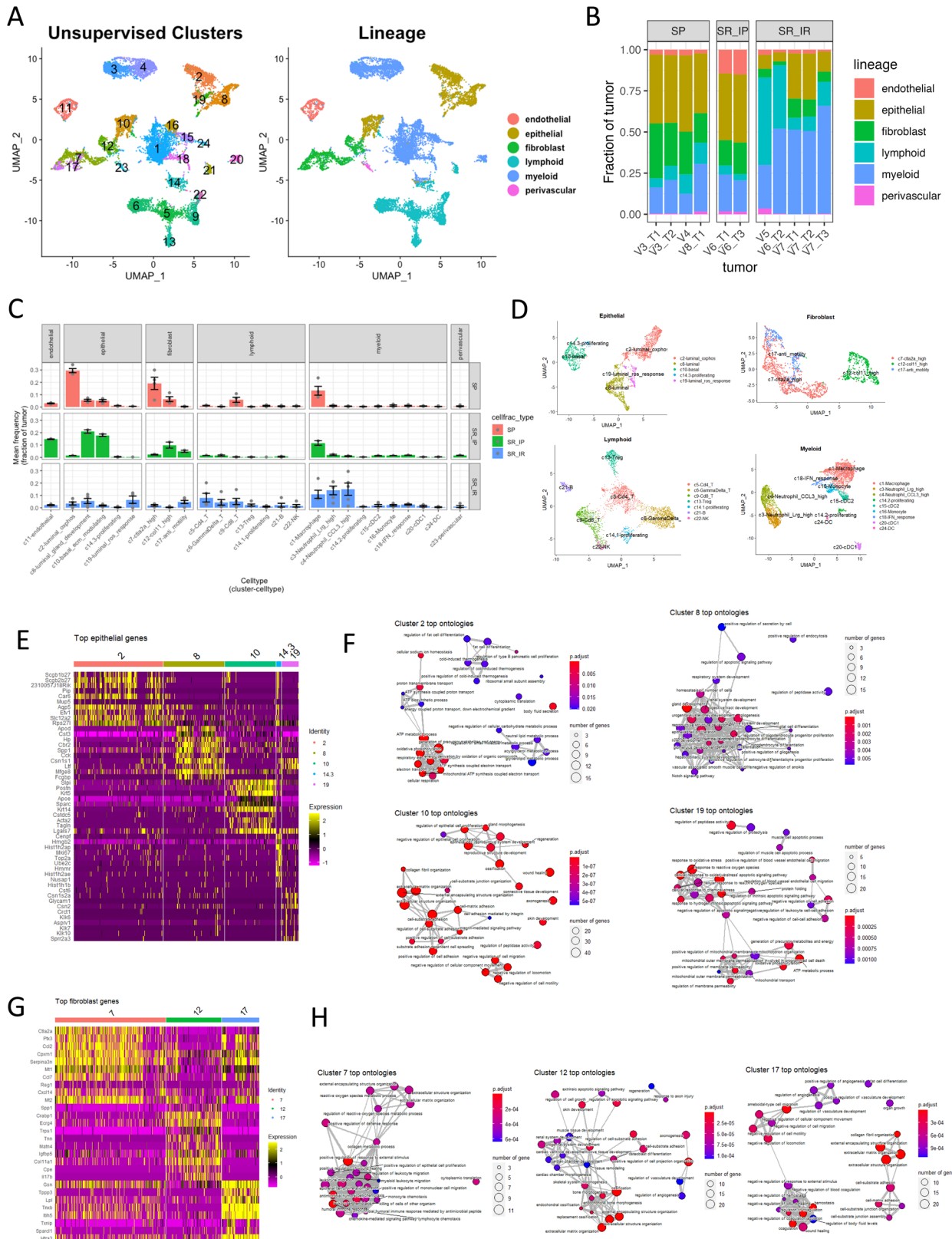

## Myc;Ptenfl tumors have shared transcriptomic signatures with human TNBC

We next used two methods in parallel to compare the single-cell transcriptomic data of our murine mammary tumors with human breast cancer in both a supervised and unsupervised manner (Fig. 7A). In the first method, we trained a cell-type classifier using public

annotated human scRNA-seq breast cancer data[74] and evaluated whether the cell states identified in human disease mapped to those found within the Myc;Ptenfl model (Fig. 7A). We trained a mixture discriminant analysis classifier[75] on scRNA-seq data from 26 primary human breast tumors[74] (Supplemental Fig. S15). A comparison of the 28 scPred assigned cell types and the 24 original clusters found in the

**Fig. 6 | Myc;Ptenfl subtypes show distinct cell types and cluster gene enrichments. A** UMAP showing scRNA-seq data from Myc;Ptenfl model. Color-coded by either lineage or unsupervised cluster. UMAP was computed on 50 iNMF factors. Lineage was manually assigned using canonical cell type markers (Source Data) for unsupervised clusters. Unsupervised clusters were identified using the Leiden algorithm to the same 50 iNMF factors (Resolution = 0.45, optimized for silhouette width, Supplemental Fig. S13B). **B** Bar-plot is showing the relative fraction of each tumor assigned to each cell lineage. Stromal-Rich (SR) and Stromal-Poor (SP) subtypes were based on initial two-class histology assigned by a pathologist and cell type fractions, which were further divided into Stromal-Rich-Immune-Rich (SR-IR) and Stromal-Rich-Immune-Poor (SR-IP). **C** Relative abundance of cell type clusters for each Myc;Ptenfl tumor subtype. Mean frequency is the arithmetic mean across all tumors within that subtype, and error bars represent SEM. **D** UMAP visualizations of individual lineages. UMAPs and clustering were computed using the same 50 factors as global analysis (Fig. 6A, Supplemental Fig. S13C). **E** Heatmap showing the top 10 uniquely upregulated genes for each epithelial cluster (min.pct = 0.1, avg_log2FC > 0.5, Bonferroni corrected $p \leq 0.05$). **F** Enrichment maps showing the top 30 enriched ontologies for each epithelial cluster visualized as a network. The size of the point indicates the number of genes within the ontology that were uniquely upregulated in that cluster. Edges connect any ontologies with a Jaccard similarity greater than 0.2 and edge width scaled to Jaccard similarity. **G** Heatmap showing the top 10 uniquely upregulated genes for each fibroblast cluster (min.pct = 0.1, avg_log2FC > 0.5, Bonferroni corrected $p \leq 0.05$). **H** Enrichment maps showing the top 30 enriched ontologies for each fibroblast cluster visualized as a network. The size of the point indicates the number of genes within the ontology that were uniquely upregulated in that cluster. Edges connect any ontologies with a Jaccard similarity greater than 0.2 and edge width scaled to Jaccard similarity.

analysis of the MycPtenfl tumors showed that most murine-derived clusters were associated with a single cell type found in human data. There was a high agreement for stromal cell types falling within the fibroblast, endothelial, perivascular, lymphoid, and myeloid lineages with an adjusted rand index (ARI) of 0.408 (Fig. 7B). Notably, cluster 7 mouse fibroblast of SP showed agreement with both human CAFs: MSC iCAF-like and myCAF-like. Where mouse fibroblast cluster 17 of SR-IR showed the most agreement with MSC iCAF-like and cluster 12 of SR-IP with myCAF-like (Fig. 7B). Epithelial cells had weaker similarity (ARI: 0.091) between Myc;Ptenfl unsupervised clusters and scPred assigned class (Fig. 7B, mouse model (MM): c10, 19, 2, 8 and Human Species (HS): Cancer, LumA/B, Her2, Basal, Progenitor), however the epithelial states in the human reference data were themselves assigned via a classifier and are therefore not intrinsic to the underlying data.

Due to low epithelial consensus between the reference human and MycPten;fl scRNA-seq data, we then directly integrated the two data sets to identify shared transcriptional states in an unsupervised fashion (Fig. 7A). Data integration and linear dimensionality reduction were performed simultaneously using Unshared INMF (UINMF)[76]. In total, 1231 shared orthologs, 1445 human-specific genes, and 268 mouse-specific genes were reduced to 50 metagene factors. The cell embeddings corresponding to these 50 metagene factors were then utilized for integrated cross-species analysis. Comparison of average UINMF embedding for each cell type identified in individual species analysis indicated that the UINMF approach performed well in learning meta-gene factors representing shared biology conserved across species (Fig. 7C). Jaccard similarity was computed to identify the overlap in the top 50 weighted genes for each factor and found that groups of factors were associated with specific cell types or lineages and shared across the two species (Supplemental Fig. S15A, B).

Unsupervised clustering of the 50 UINMF factors identified 36 integrated clusters (Supplemental Fig. S15C, D). Lineage assignments for these clusters were largely concordant with our prior analyses (Fig. 7D and Supplemental Fig. S15D, E). Notably, all 36 integrated clusters consisted of both human and mouse cells, highlighting the overlap in transcriptional signature between the Myc;Ptenfl mouse model and human breast cancer (Fig. 7D–F). Two of the three cross-species clusters that had a majority (>50%) Myc;Ptenfl cells (c31, c34) consisted mostly of neutrophils (Fig. 7D–F), which are absent in the human data set, likely due to difficulties in sequencing primary tumor neutrophils because of their high RNase content[77]. The remaining mouse-specific cluster (c25) primarily consisted of Myc;Ptenfl cells from the epi_c2_luminal-oxphos cluster (Fig. 7D–F). While this cohort of human breast scRNA-seq did not include any tumors dominated by the OXPHOS phenotype, OXPHOS-upregulated tumors have been reported in human breast cancer and are associated with metastasis and chemoresistance[40]. Overall this analysis shows that the MycPten;fl model faithfully recapitulates both stromal and malignant transcriptional programs directly observed in human disease.

## Discussion

Amplification of the *MYC* gene and loss of the *PTEN* tumor suppressor is common in human TNBC, and the majority of *PTEN* loss TNBC have a copy number gain in *MYC*, which is prognostic for poor overall survival[19–21,30]. Here we describe the generation and characterization of the Myc;Ptenfl GEMM, which was designed to simulate the molecular and biological complexity of MYC gain and PTEN loss observed in aggressive human TNBC. A major obstacle to identifying actionable targets in TNBC is the heterogeneity of the disease, both inter- and intra-tumoral, highlighting the need for robust in vivo models that recapitulate the spectrum of molecular and biological characteristics of TNBC. Our generation of the Myc;Ptenfl mouse model reveals insights into how deregulation of the *MYC* oncogene and loss of the tumor suppresser *PTEN* can cooperate in vivo to generate TNBC tumors that recapitulate the heterogeneity of human TNBC subtypes as evidenced by (1) histology illustrating similar tissue architecture and cellular morphologies, (2) immunohistology displaying the presence of a similar spectrum of diverse immune cell types, (3) RNA-seq showing the percent of similar transcriptomic signatures, (4) multiplex imaging defining similar tumor and microenvironment cell phenotypes, and (5) single-cell RNA-seq revealing the presence of multiple cancer and stromal cell populations and their gene expression profiles. Myc;Ptenfl tumors recapitulate specific elements of human TNBC tumors and tumor microenvironments, such as OXPHOS high, ROS high tumor cells, fibroblast heterogeneity including iCAF and myCAF populations, and immune low, immune suppressive, and immune high cell tumors corresponding to the spectrum of human TNBC.

An important feature of our Myc;Ptenfl TNBC model is the increased inter- and intra-tumoral heterogeneity. This is likely a consequence of the closer to the disease-relevant expression of the two copies of Rosa-driven *Myc* knocking genes, which is more similar to copy number gain in MYC seen in human breast cancer (Fig. 1B and[32]). Unlike most Myc transgenic models with strong transcriptional enhancers, such as MMTV, that can drive tumors on their own, our model requires additional oncogenic hits to drive tumorigenesis, allowing for the evolution of cellular heterogeneity, including different levels of post-translational MYC stabilization across tumor subtypes. At a broad histologic assessment, we observe two classes of Myc;Ptenfl tumors: Stromal-Rich, with several histological subgroups and 77% occurrence, and SP with 23% occurrence. The more heterogeneous group, the SR, recapitulates the human mesenchymal TNBC subtype, marked by stromal desmoplasia, lobular 60%, squamous 15%, and metaplastic 2% IDC histologies, highest expression of immune-related gene signatures and a more inflamed spatial pattern, including adjacent, peripheral and tumor-infiltrating immune cells. Gene expression analysis revealed increased hallmarks of EMT, tumor-promoting inflammatory response, and apoptosis, consistent with mesenchymal TNBC[78]. This provides a biological rationale for using the Myc;Ptenfl-SR model in drug discovery studies requiring infiltrating immune and

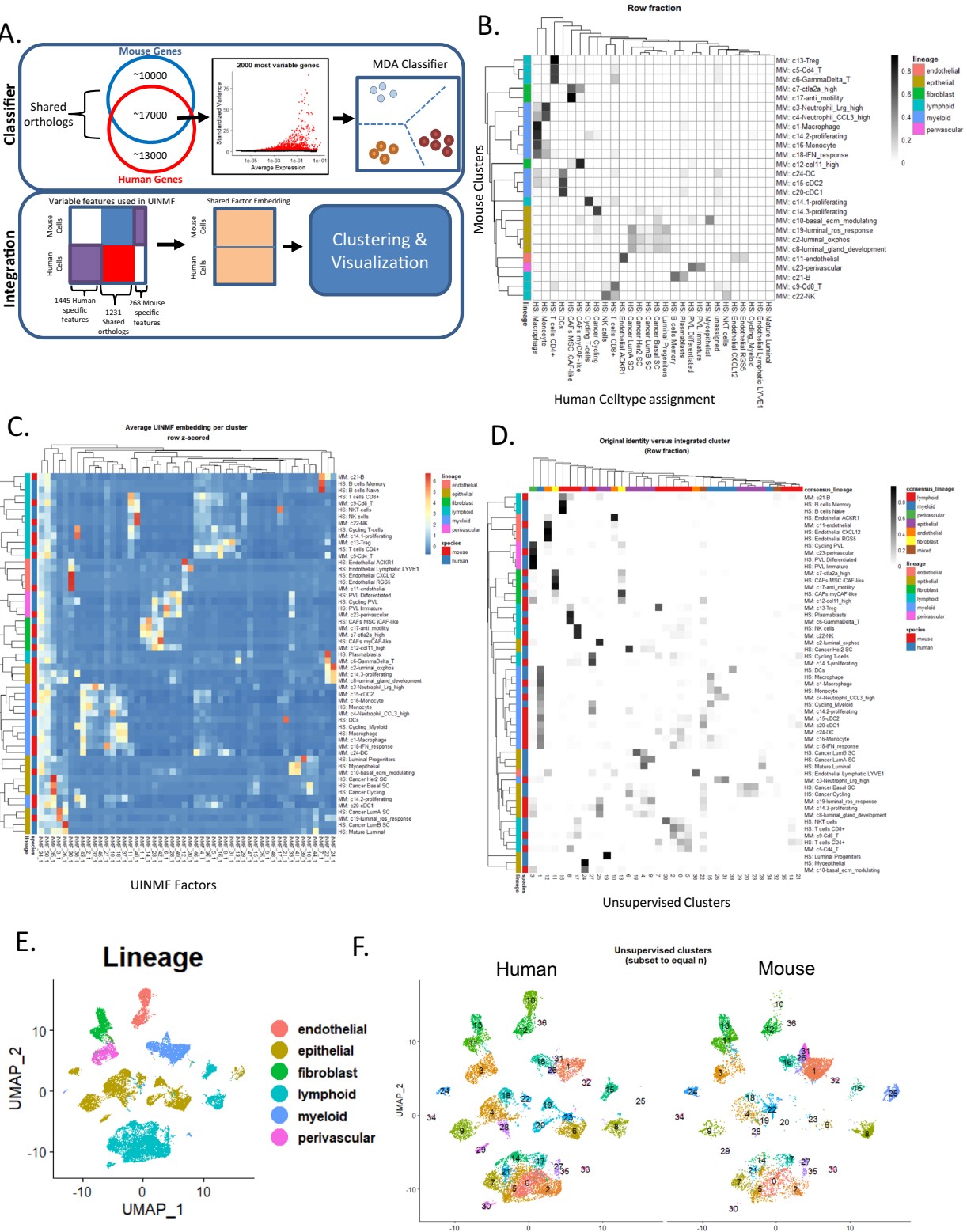

**Fig. 7 | Integration of TNBC mouse model with human breast cancer scRNA-seq.**
**A** Schematic showing classifier and UINMF integration. **B** Heatmap visualizing the relationship between Myc;Ptenfl unsupervised clusters and classifier assignment from a mixture discriminant analysis-based classifier built with scPred from human primary breast cancer scRNA-seq[74]. Counts were row-normalized to represent the fraction of each unsupervised cluster assigned to each human cell type class.
**C** Mean UINMF embedding for each cell type found in the Myc;Ptenfl or human scRNA-seq as assigned by Wu et al. **D** Original cell type identity versus integrated unsupervised cluster assignment. Counts were row normalized and represented

the proportion of each species-specific cell state that was assigned to each cross-species unsupervised cluster. **E** UMAP of UINMF integrated Myc;Ptenfl, and human scRNA-seq data. UMAP was computed from 50 UINMF factors. Lineage was assigned during the initial analysis of Myc;Ptenfl data (Supplemental Fig. 13C) (Source Data) or based on Wu et al.[74] celltype_major classification. **F** Unsupervised clusters of cross-species integrated data computed with the Louvain algorithm from 50 UINMF factors (Supplemental Fig. S15A−C) Note: Point size is increased for mouse umap.

tumor-associated stromal cells, such as immune targeting strategies. The Stromal-Poor subtype recapitulates mostly human TNBC subtypes that molecularly classify as mixed basal-like subtypes. The human data of Burstein et al. and of Ding et al. identified similar mixed subtypes in 23.7% and 33% of breast cancer samples[11,12]. SP is marked by solid invasive ductal carcinoma histology, high pS62MYC expression, a hallmark of cellular metabolism including OXPHOS, and poor prognosis. Higher MYC stability promotes an immune suppressive environment and OXPHOS signature epithelial state, which has been shown to enhance metastatic lethality and be associated with chemotherapy-resistant TNBC[79]. Therefore, the Myc;Ptenfl-SP model can provide important insights into potential mechanisms of tumor aggressiveness and therapeutic failure that will guide the future development of novel therapeutic targets in TNBC.

Interrogation of immune contexture with spatial analysis in our tumor model revealed higher total immune cells in SR, including both myeloid and lymphoid lineage cells with significant CD8 T cells in the periphery compared to SP tumors. Consistently, and similar to human TNBC, SR tumors were associated with better prognosis, whereas the SP subtype with OXPHOS metabolism, shown to be important for the production of biosynthetic intermediates necessary for rapid proliferation[40], is immune low except for macrophages and associated with poor prognosis. This type of result suggests that SP tumors would need agents to boost immune cell influx and recruitment, such as chemotherapy and Dendritic Cell agonists, whereas SR would need agents to overcome myeloid immune suppression and re-activate T cells. SP would likely also require this approach once an inflammatory response has been evoked. The ability of the Myc;Ptenfl mouse models to represent different levels of immune activity suggests that it may provide a robust platform for TNBC preclinical trials of newly developed immunotherapies. Currently, immunotherapy using checkpoint blockade has been shown to produce a long-lasting response in highly immunogenic cancers[80,81]. Although breast tumors generally are not highly immunogenic, TNBC constitutes a varied spectrum of tumors with different degrees of immunogenicity that may include a more immunogenic subtype[82]. This suggests that TNBC with a higher level of lymphocyte infiltration may be more responsive to immunotherapy[81].

There is a pressing need for pre-clinical models of breast cancer, which include the complete gamut of tumor microenvironment cell types to represent human disease and better assess cutting-edge therapeutic approaches. Our scRNA-seq analysis of the Myc;Ptenfl tumors revealed the extent of tumor cell and microenvironment heterogeneity, and we identified discrete and transcriptionally diverse populations of epithelial, fibroblast, lymphoid, and myeloid cells. Comparison with histologically assigned tumor subtypes revealed that the tumor subtype was intrinsically related to each tumor's dominant epithelial and fibroblast cluster. We found that the SP tumors were enriched for epithelial cells with high OXPHOS pathway activity as well as fibroblasts with increased expression of the immune suppressive ligand Ctla2a. These results suggest that TNBC with these signatures are likely to be poor outcomes and drug-resistant. SR tumors with low immune presence were enriched for a heterogeneous mix of epithelial cells as well as myofibroblasts. SR tumors with high immune presence uniquely had epithelial cells with high reactive oxygen species (ROS) response pathway activity, indicating crosstalk of these epithelial cells with infiltrating immune cells[83]. ROS are also able to trigger programmed cell death (PCD), leading to apoptosis[84], a pathway that is also enriched in SR immune-rich tumors. Furthermore, the SR immune-rich tumors were also enriched for an inflammatory CAF state with high expression of pathways related to immune activation and anti-motility. This comparison can be used to identify novel tumor-specific targets that may play an important role in tumor growth, progression, metastasis, and drug vulnerability. Extensive comparison with a human-trained classifier and integration across species

demonstrates that the cell types and states found in this murine model of TNBC are fully representative of human disease and share high transcriptional overlap with patient data[74].

Because the Blg-cre activation of *Myc* expression and deletion of *Pten* requires lactation (expression of Blg), the MycPten;fl model only develops tumors in female mice. While females represent >99% of breast cancer patients in humans[85], this does represent an important caveat when using the MycPten;fl model to evaluate any systems or therapeutics that are dependent on endogenous levels of male-associated androgens or other hormones. Additionally, the correlation analysis performed on human TNBC subtype signatures showed that only 1 of 13 Myc;Ptenfl tumors used for bulk RNA-seq (~8%) showed correlation to the LAR subtype, and there is less than 1% AR positive staining in each SP and SR tumors. Due to low representation, the MycPten;fl model would be an inefficient model to study the LAR subtype. However, LAR subtypes are found in a significantly smaller proportion of TNBC cases[86]. Moreover, it is well-established that AR expression is most frequently detected in ER-positive BC. In contrast, AR-positive tumors are considerably less prevalent in TNBC[87].

In summary, the development of cancer models that mimic TNBC microenvironment complexities is critical to developing effective drugs and enhancing disease understanding. This study addresses a critical need in the field by identifying a murine model that faithfully mimics human TNBC heterogeneity and establishes a foundation for future translational studies. The characteristics of our Myc;Ptenfl model provide insight into the molecular pathways involved in specific breast cancer subtypes and should serve as a platform for preclinical drug screening of heterogeneous TNBC with metastasis, including both cell-intrinsic targeted therapy strategies and the testing of immunotherapies, and combinations thereof.

## Methods

### Study approval

The research complies with all relevant ethical regulations. All protocols for mouse experiments described in this study were approved by the Oregon Health & Science University Animal Care and Use Committee protocol # IP00001014, Portland, OR.

### Antibodies

HER2 (Cell Signaling #2242, 1:50); ERa (Millipore #04-227, 1:50); PR (Abcam #ab131486, 1:1000); AR (Abcam #ab47563, 1:50); cytokeratin 5 (Abcam #ab52635, 1:100); cytokeratin 14 (Covance #PRB-155P, 1:1000); pSmad3 (Abcam #ab52903, 1:100); Laminin (Abcam #ab11575, 1:50); SMA (Abcam #ab5694, 1:100); pS62 Myc rat monoclonal 4B12;[88] Ki-67 (Abcam #15580, 1:1000); CSF-1R (Santa Cruz #sc-692, 1:500); F4/80 (Serotec A3-1, 1:200); CD11C (Cell Signaling #97585,1:100); CD4 (Cell Signaling #25229, 1:100); MHCII (eBioscience #eB14-5321, 1:100); BTK (LSBio #LS-C180161, 1:100); CD45 (BD Bioscience #550539,1:50); PDL1 (Cell Signaling #13684,1:50); CD8 (eBiosceience #14-0808082, 1:100); CD3 (Thermo #RM-9107-s, 1:300); CD207 (eBioscience #14-2073-82, 1:100); CD206 (Abcam #64693, 1:1000); B220 (BD Bioscience #550286, 1:100); RORgt (Abcam #ab207082, 1:100); Foxp3 (eBioscience #14-5773-82, 1:100); GATA3 (Abcam #ab199428, 1:100); CD11b (Abcam #ab133357,1:100); TCF1/TCF7 (Cell Signaling #2203s, 1:100); TIM3 (Cell Signaling #83882, 1:200); EOMES (Abcam #ab183991,1:1000); Granzyme B (Abcam #ab4059, 1:200); Ly6G (eBioscience #551459, 1:200); PAN Keratin (Abcam #ab27988, 1:100). CK5 (abcam, EP1601Y, 1:100); S100A6 (CST, D9F9D, 1:100); CD11c (CST, D1V9Y, 1:100); CD103(Biolegend, 2E7, 1:100); aSMA (Santa Cruz, 1A4, 1:100); EpCAM (CST, E6V8Y); CD31 (Abcam,EPR17260, 1:100); ColVI (MDBiosciences, EPR17072, 1:100); CD11b (Abcam, EPR1344, 1:100); Ki67 (CST, D3B5, 1:100); FoxP3 (Novus, NB100-39002, 1:100); Vim (CST, D21H3, 1:100); CD45 (CST, D3F8Q, 1:100); Gal3 (Biolegend, 125408, 1:100); ColIV (MDBiosciences, 203003, 1:100).

## Animal studies

Rosa-LSL-Myc mice[32] swere crossed with *Pten^{flox/flox}* (Akira Suzuki et al. Immunity 2001) and Blg-Cre mice (gift from Owen Sansom, Beatson Institute for Cancer Research, Glasgow, United Kingdom) to generate mice that express MYC and deleted PTEN in response to Cre-mediated recombination in the mammary gland. All are female and in an FVB background. Because the Blg-cre activation of Myc expression and deletion of Pten requires lactation (expression of Blg), the MycPten;fl model only develops tumors in female mice. The PTENfl and MYC;P-TENfl mice investigated in this manuscript are 100% penetrance in FVB background, and tumors were from independent animals. Tumor-bearing mice were treated with Paclitaxel at a dose of 5 mg/kg/week by intraperitoneal injection for 30 days, tumor growth was recorded every 5 days by measuring the diameter in cm. Tumor volume was calculated using the following formula: large diameter × (small diameter)$^2$/2. If a tumor impaired an animal's mobility, became ulcerated, or appeared infected, or a mouse showed hunched posture, the mouse was euthanized. According to the Institutional Animal Care and Use Committee (IACUC) protocol, the maximum allowed tumor size or burden is 2 × 2 cm. Thus, the maximum permitted tumor volume is 4000 mm$^3$. Mice will be euthanized when a single tumor is 2 cm in diameter or, in the case of multiple tumors, 1.5 cm/tumor. In our study, we remained within this permitted limit and did not exceed it. All the calculated tumor volume measurements for all mice before these were euthanized are available in the Source Data file. We followed the IACUC policy, utilizing the Carbon dioxide (CO$_2$) inhalation method of euthanasia. In this approach, compressed gas served as the source of CO$_2$ for euthanizing rodents, with the home cage serving as the euthanasia chamber. Additionally, a secondary confirmatory method of euthanasia, namely Cervical dislocation, was employed following the CO$_2$ method. Tumors were harvested and frozen for RNA and DNA analysis or embedded in paraffin for immunofluorescence or multiple immunohistochemistry staining.

The mice in our study were housed at the Oregon Health & Science University Animal Care facility, which adheres to rigorous animal welfare standards and guidelines to ensure the well-being and proper care of the animals. The housing conditions for the mice were carefully maintained as follows: Firstly, a 12-h light and 12-h dark cycle was implemented to simulate the natural light variations that mice experience in their native environment, providing them with a suitable diurnal rhythm. Secondly, the ambient temperature in the housing facility was consistently maintained within the range of 65–75 °F (-18–23 °C). This temperature range was chosen to create a comfortable and stable environment for the mice, minimizing any potential stress or discomfort. Lastly, the humidity levels in the housing facility were carefully regulated to fall between 40 and 60%. This range was considered optimal for the well-being of the mice, ensuring an appropriate level of moisture in the air without causing any excessive humidity-related issues. These housing conditions were implemented to create an environment that supports the overall welfare and health of the mice throughout the duration of the study.

## H&E staining, immunofluorescence, and immunohistochemistry

H&E staining, immunofluorescence, and immunohistochemistry were performed as described previously[32].

## Bulk RNA-sequencing and gene expression analyses

RNA was extracted from Myc;Ptenfl tumor tissue (*n* = 13) or normal mammary gland as control (*n* = 3) using Trizol (Invitrogen). RNA-Seq libraries were constructed using a NEBNext Ultra Directional RNA Library Prep Kit for Illumina (New England BioLabs) and then sequenced on an Illumina HiSeq at the OHSU Massively Parallel Sequencing shared resource. Gene expression reads per kilobase of the transcript, per million mapped reads (RPKM), were calculated for all genes. For hierarchical clustering, we performed Voom (80)

normalization on tumor samples using all genes, then reduced them to unique gene symbols and used Ward's clustering to identify tumor subgroups. Normalized count data was generated using the standard DESeq2 (v1.36.0) workflow with Variance Stabilizing Transformation[89–91]. The top 1000 genes and most variable genes were used to perform Principal Component Analysis with the "princomp" function of R package stats (v4.2.0). To normalize PCA visualization, PC embedding was scaled by dividing the embedding by the proportion of variance explained by that component. Differential expression analysis was performed with DESeq2 with LFCshrinkage via apeglm[92]. Geneset enrichment analysis (GSEA) was performed using the R package Clusterprofiler[93] (v4.4.4) with the MSigDB[39] hallmark genesets, accessed via the R package msigdbr. GSEA results are detailed in Supplemental data set 1. To perform TNBC subtype correlation analysis, we first converted the 77 subtype-associated human genes to murine genes using R package biomaRt (v2.52.0)[94,95] The 60 human genes which mapped to homologous murine genes were retained, and Spearman correlation was used to compare the converted and z-scored 60-gene TNBC subtype centroid signatures to MycPten;fl VST normalized and z-scored counts[12].

## Sequential mIHC staining and analysis

Sequential IHC was performed on 5 μm FFPE sections as previously described[43,44]. Image processing and cell quantification were performed as previously described[96,97].

## Mouse and human TMAs

The mouse Myc;Ptenfl TNBC TMA was generated by marking tumor regions of interest on FFPE blocks and punching 1.5 mm cores using TMA Master II (3DHistech, Hungary) for drilling recipient block and MTA-1 (Estigen Tissue Science, Estonia) for tissue coring. The tumors in the human TMAs were determined to be TNBC by pathologist (MES) evaluation of IHC staining for ER, PR, and HER2. The human TMAs are comprised of 60% TNBC, 30% ER+, and 10% HER2+ disease. TMA101 and TMA11-4-09 were constructed from surgically resected primary tumor samples from patients with breast cancer diagnosed at Vanderbilt University Medical Center. One millimeter tumor core (two per surgical specimen) were punched from representative areas containing invasive carcinoma selected by a pathologist. Clinical and pathological data were retrieved from medical records under institutionally approved protocols, IRB# 030747 and 130916, for patients in TMA101 and TMA11-4-09, respectively. Participants gave informed consent to participate in the repository and were not compensated for participation. Informed written consent was obtained and received from all the patients involved in this study.

## Mouse and human TNBC TMA H&E histologic/morphologic analysis

Pipeline for generating morphological feature representation using a variational autoencoder (VAE) to compare tissue microarray (TMA) from our TNBC mouse model with TMA from Human TNBC. First, raw Hematoxylin & Eosin (H&E) stained TMA images (92 TMA cores from mice TMAs and 172 cores from human TMAs) were pre-processed to account for between-sample intensity variation. H&E pixel intensities were normalized using the Reinhard method[98], where background pixels are excluded from intensity distributions. One subset of human cores was used as the target distribution to which each other dataset was normalized[99]. For parameter selection and optimization, we explored H&E tile sizes of 256 × 256, 512 × 512, 1024 × 1024, 2048 × 2048 (pixels), and latent feature vector dimensions of 32, 64, 128, 256, and 512 to identify meaningful representations of the image dataset. We found that 1024 × 1024 tile size and a feature vector dimension of 64 captured meaningful histological features based on visual evaluation and yielded the lowest reconstruction losses. Tiles from both mice and human TMAs are used to train a VAE, then a latent encoding vector is computed

for each tile. Tiles are compared using UMAP embedding and k-means clustering analysis of the latent features. Density functions for all human and mouse tiles are calculated within the 2-dimensional UMAP space to visually compare overlap in embedding space. K-means clusters ($n = 8$) are computed using latent features and projected into UMAP space for visualization. The relative abundance of human and mouse TMAs are calculated for each cluster using the ratio of tiles in a cluster to total tiles from the given TMA source.

## Cyclic multiplexed-immunofluorescence (cmIF) and single-cell multiplexed analysis

Cyclic immunofluorescence, image processing, and Cyclic IF analysis were performed as previously described[64]. Antibody order was chosen to minimize known artifacts, including channel cross-talk and incomplete quenching (see Source Data for panel order and vendor information). Specifically, antibodies that stain the same cell in the same cellular compartment were positioned in non-adjacent rounds and channels. We used the free and open-source mplexable pipeline for image registration, quality control, single-cell segmentation, and feature extraction. The mplexable image processing pipeline parameters are available as a jupyter notebook at "https://github.com/engjen/MYC-PTENfl-mouse/blob/main/20210707_RS-mTMA-20220119_mpleximage_data_extraction_pipeline_bue.ipynb". As specified in the pipeline notebook, we used a scaled autofluorescence subtraction strategy, using round zero (R0) autofluorescence images and round five quenched (R5Q) autofluorescence images to linearly interpolate and subtract autofluorescence that is changing over rounds of CyCIF (see ref. 64). We extracted the pixel intensity values from autofluorescence subtracted 16-bit images using nucleus and cytoplasm segmentation masks to calculate the mean pixel intensity of each single subcellular compartment. For each marker, we selected the nuclear or cytoplasmic single-cell mean intensity for analysis depending on the expected intracellular distribution of that marker. Each marker was validated for specificity and signal-to-background ratio (SBR) through a visual review of the images. Twenty markers showing good specificity and SBR were selected for downstream analysis, including cluster analysis. In addition to marker QC, each tissue core was inspected, and areas of floating tissue that created bright imaging artifacts and air bubbles that created dark artifacts were manually circled using the Napari image viewer and excluded. Additionally, two percent of cells were filtered from analysis due to tissue loss (i.e., they were negative for DAPI staining after eight rounds of staining). Background subtraction was performed on markers with high background: CD31, CD45, CD8, ColIV, FoxP3, CD103, CD11b and CD11c. Background subtracted data were clustered with scanpy[100]. Two morphology features (nuclear area and nuclear eccentricity) and 20 markers were used for clustering. A Umap embedding was generated using 15 neighbors, and the Leiden algorithm was used for unsupervised clustering. We tested three resolution parameters for the Leiden algorithm (0.4, 0.5, and 0.6). We performed clustering and visualized each cell type cluster's spatial distribution in the tissue using the jupyter notebook available here: https://github.com/engjen/MYC-PTENfl-mouse/blob/main/20211027_RS-TMA_cluster.ipynb. We then visualized the spatial distribution of markers in the images using the Napari image viewer to load the multichannel images available here: "https://www.synapse.org/#!Synapse:syn51314365/files/" and the code available here: "https://github.com/engjen/MYC-PTENfl-mouse/blob/main/20211022_RS-TMAs_napari.py". The final resolution (0.6), resulting in 20 cell types, was selected based on a visual examination of clustering results overlaid on images. Inspection of cluster results in images revealed that 3 of the clusters were due to imaging artifacts and were excluded. The remaining clusters were evaluated on the images and annotated. Endothelial cells were separated from mixed endothelial/immune and endothelial/fibroblast clusters by manually gating based on CD31 expression.

We then performed manual gating to verify our annotated-cluster cell type calling. A threshold was set for each gating marker based on positive pixel patterns in images. Endothelial cells were defined as CD31+. Epithelial cells were positive for 1 or more of Ecad, EpCAM, CK5, and CD31−. Immune cells were CD45+ CD31− and epithelial marker negative. Stromal cells were all non-endothelial, non-epithelial, non-immune segmented nuclei. Three cores had pMYC-positive tumor cells negative for Ecad, EpCAM, and CK5 (I11, G09, and H11). In these tissues, tumor cells were defined as any cells negative for all stromal and immune-gating markers (CD31, CD45, Vim, aSMA, ColIV, ColIV, Gal3).

Cell frequencies were calculated for gated epithelial, immune, and stromal cells in each tissue core. Endothelial cells were rare and added to "stromal" cells for subtyping. We tested four resolution parameters of the Leiden algorithm (0.2, 0.3, 0.4, and 0.5) to cluster tissues based on cell frequency. We chose resolution 0.5, resulting in 6 clusters, because this captured the histological subtypes. Specifically, cluster 5, the highest in epithelial cells (or stroma poor+, SP+), was significantly correlated with the SP histological subtype based on H&E evaluation (chi-squared $p = 2.2e{-}6$). We collapsed the 6 clusters into three based on the correlation between clusters with hierarchical clustering (Supplementary Fig. S8D). Clusters were annotated as Stromal-Rich, immune-rich (clusters 2 and 4), SR, IP (cluster 0), and SP subtypes (clusters 1, 3, and 5). Most of the cyclic IF SP samples were also stroma poor by histology (6/9). Discrepancies between CyCIF subtypes and histology subtypes were examined (Supplemental Fig. S16) and found to be a result of tumor-adjacent stroma, lymphocytes in the tumor core, or significant nuclei-free areas of tissue.

To compare the expression of markers in each subtype, mean marker expression was calculated in each tissue and compartment (epithelial or stromal, i.e., immune, endothelial, and non-immune stromal cells). Distributions were visualized as boxplots, and the Kruskal Wallis $H$ test or Mann Whitney $U$ test, implemented in scipy[101], were used to test for significant differences in the median expression of markers between groups. To compare detailed cell types (14 annotated cell types from Leiden clustering, described above) between subtypes, the frequency of each cell type in each subtype was calculated and displayed as a bar plot. Samples were also clustered hierarchically based on the $z$-score of cell abundances.

For human tissue samples, we obtained a publicly available multiplex ion beam imaging (MIBI) dataset[65] at https://github.com/aalokpatwa/rasp-mibi. The MIBI images were segmented, and single-cell intensity was extracted by Patwa et al.[65]. The values represent units related to the time-of-flight mass spectrometry detector reported by Patwa et al.[65]. We used scanpy for single-cell clustering. Thirty-two markers were used for generating a Umap embedding with 15 neighbors. Unsupervised clustering of cells in the MIBI dataset was performed with the Leiden algorithm using the same resolution parameter as with the CyCIF dataset (resolution = 0.6), resulting in 24 cell types. The code for clustering of MIBI cell types, including visualization of each cluster's spatial distribution in the tissue, is available here: "https://github.com/engjen/MYC-PTENfl-mouse/blob/main/20211109_MIBI_cluster.ipynb". The cell type clusters were annotated as epithelial, immune, or stromal. Inspection of cluster results versus images revealed that some clusters contained mixed immune and stromal cells. Therefore, epithelial clusters were used to define epithelial cells, and CD45 and CD31 were used to manually gate immune and endothelial cells within the stromal/immune clusters. Tissues were clustered on cell type frequencies with the Leiden algorithm, resolution = 0.1, resulting in 3 subtypes similar to the mouse data, annotated as SR, IR (cluster 0), SP (cluster 1), and SR, IP (cluster 2). Mean marker expression was calculated in each tissue and compartment, visualized, and statistically evaluated as described for mouse data above. For survival analysis, the lifelines[102] python software was used. Kaplan-Meier estimates were generated and plotted for overall survival. The log-rank test was used to test for significant survival differences between the subtypes.

## scRNA library preparation and sequencing

Single-cell suspensions of 11 Myc;Ptenfl tumors from 6 mice were obtained by enzymatic digestion. Tissue was manually minced using scissors, followed by a 30–60 min enzymatic digestion with 2.0 mg/ml collagenase A (Roche), 1.0 mg/ml Hyaluronidase (Worthington), and 50 U/ml DNase I (Roche) in serum-free Dulbecco's modified eagles medium (DMEM) (Invitrogen) and Rock inhibitor at 37 °C using continuous stirring conditions. Single-cell suspensions from tumor digests were prepared by passing tissue through 40-mm nylon strainers (BD Biosciences). Single-cell suspensions from individual tumors were then labeled with hashtag oligonucleotides following the manufacturer's protocol (TotalSeq B0301–B0306, Biolegend). Each individual tumor sample was counted and then pooled at an equal cell ratio before being split into two replicates for library preparation with the Chromium Single Cell 3′ V3 (10× Genomics) following the manufacturer's protocol with a targeted recovery of 20,000 cells per library. Libraries were sequenced on an Illumina NovaSeq. BCL files were converted to fastq format with bcl2fastq2 (Illumina) and then aligned to mouse genome build mm10-2020-A (10× Genomics) using Cellranger (10× Genomics, version 6.0.2).

## TotalSeq FVB compatible antibody for cell hashing

The BioLegend TotalSeq B0301-B0306 was used, containing CD45 (clone 30-F11) and MHC-I (clone M1/42). The M1/42 clone (anti-MHC I) is reported to recognize cells from C57BL/6 (B6) mice, which have the H-2b haplotype, but it was not tested for cells from FVB mice, which have the H-2q haplotype. To test whether M1/42 binds FVB poorly or not at all, we stained both strains with fluorophore-conjugated CD45 (clone 30-F11) and MHC-I (clone M1/42) (Supplemental Fig. S11C). Single-cell suspensions were stained in FACS buffer (1× PBS, with 2% FBS and 0.1% NaN3) at 4 °C, for 20 min, in the dark. Antibodies used are as follows: CD45 (clone 30-F11) and MHC-I (clone M1/42), EpCAM (clone G8.8) and TER-119 (clone TER-119). Cells were run on a BD FACS Symphony A5 (BD Biosciences). Data were analyzed using FlowJo 10.6 (Tree Star, Inc., RRID:SCR_008520). The M1/42 clone stained all B6 cells. CD45+ FVB stained less brightly than B6 cells but were clearly MHC I+. Some FVB cells were not stained, but of the MHC I negative cells, nearly all were TER-119+, indicating they are red blood cells that are expected to lack MHC I staining. Thus, M1/42 appears to cross-react to FVB cells, with reduced staining intensity but was sufficiently reactive to use TotalSeq (Biolegend) reagents on FVB cells (Supplemental Fig. S11C).

## scRNA-seq data processing

The R package SoupX was used to load the UMI gene count matrix for each library and correct for ambient mRNA contamination using default parameters (Young and Behjati, 2020). The corrected UMI gene count matrix was then converted to Seurat Object format using Seurat (v4.1.0), and the paired HTO count matrix was added. HTO demultiplexing was performed using the *HTODemux* function of Seurat with parameters: (kfunc = 'clara', positive.quantile = 0.95) (Satija et al., 2015; Butler et al., 2018; Stuart et al., 2019; Hao et al., 2021). Doublets were identified within each library using the R package DoubletFinder (v2.0.3) with a presumed Poisson doublet rate of 0.075 and 10 principal components(McGinnis, Murrow, and Gartner, 2019). Only cells with greater than 250 unique genes expressed, less than 25% mitochondrial RNA, and assigned as a 'Singlet' via DoubletFinder were retained for analysis.

## Myc;Ptenfl scRNA-seq normalization, integration, clustering, and differential expression analysis

UMI counts were log normalized and scaled without centering using the R package Seurat (v4.1.0). The top 2000 variable features were identified using the VST method using the R package Seurat. iNMF integration was performed directly on the Seurat object accessing the rliger package (v1.0.0) via the SeuratWrapper package (v0.3.0). OptimizeALS was run with the parameters: ($k$ = 50, lambda = 5, nrep = 5, split.by = 'library_id'). RunQuantileNorm was performed with the same split.by setting. UMAP visualization was performed using the resultant 50 iNMF factors. Unsupervised clustering was performed using the Leiden algorithm with a resolution of 0.45, which corresponded to the point of diminished returns for increased silhouette width as estimated by the 'approxSilhouette' function of the R package Bluster (v1.2.1).

## scRNA-seq differential expression and enrichment of gene ontology

Differential expression analysis was performed using the Wilcoxon rank-sum test via Seurat's 'FindMarkers' function, and significantly differentially, genes for any cluster had at least a log2 fold-change of 0.5, and Bonferroni corrected $p$ value below 0.05. The R package ClusterProfiler (v4.0.5) was used to identify enriched gene ontologies from significantly upregulated genes with parameters: (ont = 'ALL', pAdjustMethod = 'BH', pvalueCutoff = 0.01, qvalueCutoff = 0.05)[93].

## Cross-species classifier

Orthologous genes found in both the Wu et al. dataset and Myc;Ptenfl scRNA-seq data were found using the 'convert_mouse_to_human_symbols' function in NicheNetR (v1.0.0). The Wu dataset was subset to only include orthologous features, and a classifier was trained using the mixture discriminant analysis (MDA) model of scPred (v1.9.2). The classifier was then applied to the Myc;Ptenfl data with a threshold of 0.55. Adjusted Rand Index was computed with the R package aricode (v1.0.0) and used to compare the original labels derived from unsupervised clustering and transferred classes from scPred classifier.

## Cross-species scRNA-seq data integration and analysis

Orthologous genes found in both the Wu et al. dataset and Myc;Ptenfl scRNA-seq data were found using the 'convert_mouse_to_human_symbols' function in NicheNetR (v1.0.0), and the mouse features were updated to human symbols. The UMI count matrices for mouse and human datasets were loaded with rliger (v1.0.0) and log normalized. Variable genes were found with the 'selectGenes' function of rliger using parameters:(var.thres = 0.3, unshared = TRUE, unshared.thresh = 0.3, unshared.datasets = list(1,2)), and 1231 shared features, 268 mouse features and 1445 human features were selected. Variable features were scaled without centering in rliger, and then the optimizeALS function was run to perform matrix factorization using parameters: (lambda = 5, use.unshared = TRUE, thresh = 1e-10, k = 50, nrep = 5). The rliger 'quantile_norm' function was then used to build the shared factor graph and normalize with parameter: (ref_dataset = 'human'). The liger object was then converted to Seurat format with the rliger function 'ligerToSeurat.' The cell embeddings from all 50 UINMF factors were used for UMAP embedding and unsupervised clustering. Unsupervised clustering was performed using the Louvain algorithm as implemented in Seurat, and a resolution of 0.6 was selected as it optimized for maximum mean silhouette width as estimated with the 'approxSilhouette' function of the R package Bluster (v1.2.1). Unsupervised clusters were assigned lineage if there was at least 80% agreement of prior lineage annotation of the cluster's constitutive cells, otherwise they were labeled 'mixed.'

## Statistics

Spearman correlation coefficient was used to assess correlations of percentages and densities among tumor sample lineages. Unsupervised hierarchical clustering was performed with Ward's minimum variance method ("hclust" from "R"). All statistical calculations were performed by R software, version 3.5.2 (https://www.r-project.org). $p < 0.05$ was considered statistically significant.

Statistical analysis was performed using GraphPad Prism software. Measurements are presented with sample *n* and mean ± SD or SEM as indicated in figure legends. An unpaired two-tailed Student's *t* test was used throughout to compare the two groups. A base *p* value of <0.05 was considered statistically significant.

**Reporting summary**

Further information on research design is available in the Nature Portfolio Reporting Summary linked to this article.

## Data availability

The bulk RNA-sequencing data generated in this study have been deposited in the Gene Expression Omnibus (GEO) database under accession number GSE215071. The mIHC data generated in this study are provided in the Source Data file. The row data of the morphological features of Myc;Ptenfl tumors generated in this study are provided in the Source Data file. The data used for Cyclic Multiplexed-Immunofluorescence in this study is available at [https://github.com/engjen/MYC-PTENfl-mouse] The MIBI publicly available data used in this study are available through the GitHub[65]. The single-cell RNA-sequencing data generated in this study have been deposited in the Gene Expression Omnibus (GEO) database under accession number GSE215070. The scRNA-seq publicly available data used in this study are available through the Gene Expression Omnibus under accession number GSE176078[74]. Source data are provided in this paper. The Single Source Data file is available, containing the following datasets: Bulk_msigdb_GSEA, Mice Tumor Volume (mm[3]), mIHC Antibodies, mTMA-Markers, Myc;Ptenfl mice TMA histologic features, and Cell type_markers used in the scRNA-seq analysis. Source data are provided in this paper.

## Code availability

RNA-seq analysis code generated for this study is available at: https://github.com/zdoha/MycPtenfl-TNBC-model. Variational autoencoder (VAE) code for Histologic/morphologic analysis generated for this study is available at: https://github.com/schaugf/ImageVAE. The Cyclic Multiplexed-Immunofluorescence analysis code generated for this study is available at https://github.com/engjen/MYC-PTENfl-mouse. R code for analysis of scRNA-seq data generated for this study can be found at https://github.com/HeiserLab/NatureComms_MycPtenAtlas.

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

## Acknowledgements

We would like to acknowledge Karyn Taylor for her invaluable support in timely reagent ordering and logistical arrangements throughout this study. RNA sequencing was performed in the Massively Parallel Sequencing Shared Resource at OHSU. Some of the embedding and sectioning of tissues was performed by the OHSU Histopathology Shared Resource, and some of the imaging was performed by the Advanced Light Microscopy Core. The OHSU Shared Resources are supported by the Knight Cancer Institute through NIH P30 CA69533. This project was funded in part by the OHSU Knight Cancer Institute. Funding for these studies includes U54 CA209988 to JWG and RCS, R01 CA186241, R01 CA129040, U01 CA224012, Prospect Creek Foundation, and Brenden-Colson Center Foundation to RCS; NIH P50 CA098131 and Komen SAB210301 to J.A.P. NLC acknowledges support from the NIH through training grant NCI T32CA254888.

## Author contributions

Z.O.D., X.W. and R.C.S. designed the study. M.M. examined the METABRIC dataset of primary breast cancer. X.W. and Z.O.D. performed all animal studies, and Z.O.D. monitored and prepared mouse tumors for single-cell suspensions. Z.O.D., N.L.C. and C.P. performed the bulk RNA-Seq analysis and clustering, and Z.O.D. performed GSEA analyses and human correlation. Z.O.D. and X.W. performed the IHC staining and conducted an in vivo study for Paclitaxel treatment. L.T., E.N.K. and Y.H.C. performed the tissue microarray (TMA) histological and morphological analysis and write-up. Z.O.D. analyzed and compiled human TNBC histologic features frequency in Myc;Ptenfl tumor subtypes' tissue microarray. X.W., Z.O.D., C.B., N.K. and L.M.C. performed or supported the mIHC staining and analysis; Z.O.D. performed the data presentation. J.A.P., M.E.S., T.W. and X.W. generated the human and mouse tissue microarrays; J.W.G. and K.C. directed CycIF workflow; Z.T. and D.B. performed antibody validation for the CycIF staining; S.K., E.B. and J.E. performed CycIF staining, image processing, and multiplex imaging analysis for the TMA. M.M., A.K., X.W. and Z.O.D. performed or supported the flow cytometry analysis. Z.O.D., N.L.C., C.J.D., X.L., L.M.H. and G.M. performed or supported the scRNA library preparation and analysis. ZOD conducted optimization and implementation of a disassociation protocol for mammary tumors to obtain a highly viable single-cell suspension and performed the hashtag oligonucleotides staining for library preparation for the Chromium Single Cell 3' V3. N.L.C. performed the scRNA-Seq analysis, clustering, and cross-species scRNA-seq data integration and write-up. M.D. provided grant support and discussion for this study, while E.M.L. contributed to data interpretation. Z.O.D. and R.C.S. wrote the paper. All authors reviewed and edited the paper.

## Competing interests

Declaration of interests: Rosalie C. Sears: Consultant: Novartis Pharmaceutical, Larkspur Biosciences. Scientific Advisory Board: RAPPTA Therapeutics. Sponsored Research Support: Cardiff Oncology, Astra Zeneca Partner of Choice grant award. Gordon Mills: SAB/Consultant: Amphista, Astex, AstraZeneca, BlueDot, Chrysallis Biotechnology, Ellipses Pharma, ImmunoMET, Infinity, Ionis, Leapfrog Bio, Lilly, Medacorp, Nanostring, Nuvectis, PDX Pharmaceuticals, Qureator, Roche, Signalchem Lifesciences, Tarveda, Turbine, Zentalis Pharmaceuticals. Stock/Options/Financial: Bluedot, Catena Pharmaceuticals, Immuno-Met, Nuvectis, SignalChem, Tarveda, Turbine, Licensed Technology, HRD assay to Myriad Genetics, DSP patents with Nanostring. Sponsored research: AstraZeneca, Nanostring Center of Excellence, Ionis (Provision of tool compounds). The title and ID number for patents; Nanostring Simultaneous quantification of gene expression in a user-defined region of 10,640,816 a cross-sectioned tissue, Simultaneous quantification of a plurality of proteins in a user-defined region of 10,501,777 a cross-sectioned tissue, Myriad Methods and materials for assessing loss of heterozygosity 10,612,098. Lisa M. Coussens reports consulting services for Cell Signaling Technologies, AbbVie, the Susan G Komen

Foundation, and Shasqi, received reagent and/or research support from Cell Signaling Technologies, Syndax Pharmaceuticals, ZelBio Inc., Hibercell Inc., and Acerta Pharma, and has participated in advisory boards for Pharmacyclics, Syndax, Carisma, Verseau, CytomX, Kineta, Hibercell, Cell Signaling Technologies, Alkermes, Zymeworks, Genenta Sciences, Pio Therapeutics Pty Ltd., PDX Pharmaceuticals, the AstraZeneca Partner of Choice Network, the Lustgarten Foundation, and the NIH/NCI-Frederick National Laboratory Advisory Committee. The remaining authors declare no competing interests.

## Additional information

[1]Department of Molecular and Medical Genetics, Oregon Health & Science University, Portland, OR, USA. [2]Department of medical laboratory technology, Taibah University, Al-Madinah al-Munawwarah, Saudi Arabia. [3]Department of Biomedical Engineering, Oregon Health & Science University, Portland, OR, USA. [4]OHSU Center for Spatial Systems Biomedicine, Oregon Health & Science University, Portland, OR, USA. [5]Brenden-Colson Center for Pancreatic Care, Oregon Health & Science University, Portland, OR, USA. [6]Department of Molecular Microbiology and Immunology, Oregon Health and Science University, Portland, OR, USA. [7]Department of Cell, Developmental & Cancer Biology, Oregon Health and Science University, Portland, OR, USA. [8]Division of Oncologic Sciences, Oregon Health and Science University, Portland, OR, USA. [9]Department of Biochemistry, Vanderbilt University Medical Center, Nashville, TN, USA. [10]Vanderbilt-Ingram Cancer Center, Vanderbilt University Medical Center, Nashville, TN, USA. [11]Department of Pathology, Microbiology, and Immunology, Vanderbilt University Medical Center, Nashville, TN, USA. [12]Knight Cancer Institute, Oregon Health & Science University, Portland, OR, USA. [13]These authors contributed equally: Zinab O. Doha, Xiaoyan Wang. ✉e-mail: searsr@ohsu.edu

