## [Peer Review File · Nature Communications]

MYC Deregulation and PTEN Loss Model Tumor and Stromal Heterogeneity of Aggressive Triple-Negative Breast CancerReviewers' Comments:

Reviewer #1:

Remarks to the Author:

Doha et al developed a novel genetically engineered mouse model with amplification of the oncogene MYC and deletion of the tumor suppressor PTEN, two genetic aberrations that are commonly co-observed in human triple negative breast cancer. Using several omic based analyses, such as bulk and single-cell RNA sequencing, multiplex immunohistochemistry, and single-cell image-based spatial analysis, they performed an extensive characterization of their model. Importantly, they show through comparison with human tumors that their model mimics the inter- and intra-tumoral and microenvironmental heterogeneity of human triple negative breast cancer, something that is missing from most other genetically engineered mouse models of this disease.

The deep and thorough characterization of the Myc;Ptenfl mouse model will provide a significant resource to the triple negative breast cancer community for helping to better understand the biology of these tumors. The model itself will provide an important tool for identifying targeted therapeutics for these patients, something that remains and important unmet clinical need.

This manuscript is well written, and the data provided is extensive, comprehensive, and well analyzed, something that is incredibly useful for someone who might be deciding how best to use this mouse model in future studies.

Specific Comments

1. There are a few genetically engineered mouse models that develop mammary tumors through overexpression of Myc alone (e.g., MMTV-Myc and WAP-Myc). Could you comment more on why the mice with only the deregulated Myc did not develop mammary tumors by 24 weeks after Blg-Cre activation? If these mice are followed out longer than 24 weeks, is there evidence that they would eventually develop tumors? Do you have a hypothesis for why your Myc only model behaves differently than these other models?
2. The y-axis for Figure 1F goes from 0-150%. I would suggest changing this figure, so the y-axis is 0-100%.
3. In Figure 2F, the authors claim that "mice with SR tumors were sensitive to Paclitaxel". To support this claim, the authors should show untreated tumor growth curves for both SR and SP tumors and perform a statistical analysis to show treatment with Paclitaxel reduces tumor growth.
4. From the methods, it appears that SR and SP groupings for Figure 2F were determined post-treatment. Is it possible that treatment with Paclitaxel influences these groupings? For example, is it possible that Paclitaxel responsive SP tumors appear SR post-treatment and are incorrectly grouped?

Reviewer #2:

Remarks to the Author:

In this work, Doha et al. developed a mouse model of triple-negative breast cancer (TNBC) by mimicking two common TNBC mutations. A number of molecular experiments, including single-cell RNA sequencing and multiplexed immunohistochemistry, were conducted by the authors to demonstrate that this genetic mouse model replicates the molecular characteristics of human TNBC subtypes. This reviewer believes that this model would be extremely valuable for testing newly developed therapies. Overall, the manuscript is well organized, and the analysis is well supported by statistical data. I recommend the publication of this manuscript after addressing the following points.

1. Although the authors have already explored this topic in some detail in the publication, it would be helpful for future researchers who plan to use this model to provide a concise overview of how it differs from human TNBC. What are the limiting factors of the model? Which human TNBC molecular characteristics or subtypes were found to be missing from this model?

Considering the significance of multiplexed imaging in this work, the corresponding methods portion is relatively concise. The authors stated that multiplexed imaging was performed by following the method described by Eng et al. (Communications Biology 5, 438, 2022).

2. Eng et al. reported that surgically sectioned tissue exhibits more cell loss during the CyCIF process. Even after five rounds, tissue prepared by surgical section had a cell loss of around 10%. Please indicate how the mouse tumor tissue was obtained as well as the cell loss percentage.

3. The reviewer is impressed by the resemblance between the human expression data acquired by MIBI and the mouse model data acquired by CyCIF. These two imaging methods have distinct advantages and disadvantages. CyCIF images may include a high level of autofluorescence, whereas MIBI images do not. Eng et al. demonstrated that the level of autofluorescence in the AF488 channel varies based on various parameters and the way in which it may be removed theoretically and physically. For potential readers, it would be beneficial to specify how autofluorescence was eliminated in each round and channel in the CyCIF imaging.

4. As noted by Eng et al., the order of antibodies influences their observed signal intensity. This reviewer is impressed by the similarities between the human- and mouse-related data depicted in Figure 5. Researchers who wish to conduct similar work will benefit from details on how the antibody orders were chosen and how the signal-to-noise ratio (SBR) measurement was validated.

5. How were the dynamic ranges of the protein expression levels calculated when comparing the MIBI and CyCIF images? These two techniques have distinct dynamic ranges. Please explain how the absolute values of the protein expression levels depicted in figures 5g, 5h, 5j, and 5k were assessed and extracted from the raw pictures.

6. In multiple analyses, the authors used different resolutions in the Leiden algorithms for the unsupervised clustering. Please specify how the resolution was determined for such clustering would be helpful for readers who would like to replicate this work.

Reviewer #3:

Remarks to the Author:

This manuscript by Doha et al. developed a TNBC murine model and performed bioinformatic analysis on these Myc;Ptenfl tumors to profile human TNBC. This study of TNBC could provide a valuable tool for the field to understand the microenvironment in TNBC patients and might have potential clinical significance.

However, a few aspects of the analysis might be improved to strengthen this manuscript and clarify the narrative.

1. In the RNA-seq analysis, please consider moving the PCA plot to the main figure to show the distinctive clustering between SR and SP samples. Then, in both Supp Figure 2D and main Figure 2H,I, it seems that some SP sample clusters with SR. Maybe the authors would want to elaborate on the nature of the outlier sample due to the relatively small sample size for each histology. In the heatmap, instead of showing all the genes without labeling, it might be clearer to just show the top genes in the principle components and perform a GO analysis using those genes.

2. In Supp Figure 7A, it is shown that Ecad is only expressed in the cluster on the right. If Ecad is used for manual gating of epithelial cells, why are epithelial cells also seen overlapping with stromal or

immune cells in Supp Figure 7B? Please comment on the UMAP clustering patterns seen in Supp Figure 7B. The batch effect removal seems to be overdone such that the distinction between the three major populations is now missing.

3. For Figure 6, to better illustrate the difference in cell composition between SR and SP, maybe the authors should consider showing a UMAP by the two phenotypes.

4. Most of Supp Figure S8 panels are not referenced in the manuscript, please fix. For a small dataset as is described in Supp Figure S8, please provide more rationale for clustering resolution. Panel 3 seems to be overclustering.

5. Please adjust the layout of Figure 5, some panels are overlapping.

Point-by-point response to reviewer comments:

We would like to express our gratitude to the reviewers for their thorough and insightful comments on our manuscript. Addressing each comment has strengthened our study and clarified our manuscript. Our point-by-point responses are below, and updates in the manuscript are in colored text for ease of review.

Reviewer #1, expertise in breast cancer models, TNBC and single cell sequencing (Remarks to the Author):

Doha et al developed a novel genetically engineered mouse model with amplification of the oncogene MYC and deletion of the tumor suppressor PTEN, two genetic aberrations that are commonly co-observed in human triple negative breast cancer. Using several omic based analyses, such as bulk and single-cell RNA sequencing, multiplex immunohistochemistry, and single-cell image-based spatial analysis, they performed an extensive characterization of their model. Importantly, they show through comparison with human tumors that their model mimics the inter- and intra-tumoral and microenvironmental heterogeneity of human triple negative breast cancer, something that is missing from most other genetically engineered mouse models of this disease.

The deep and thorough characterization of the Myc;Ptenfl mouse model will provide a significant resource to the triple negative breast cancer community for helping to better understand the biology of these tumors. The model itself will provide an important tool for identifying targeted therapeutics for these patients, something that remains and important unmet clinical need.

This manuscript is well written, and the data provided is extensive, comprehensive, and well analyzed, something that is incredibly useful for someone who might be deciding how best to use this mouse model in future studies.

Specific Comments

1. There are a few genetically engineered mouse models that develop mammary tumors through overexpression of Myc alone (e.g., MMTV-Myc and WAP-Myc). Could you comment more on why the mice with only the deregulated Myc did not develop mammary tumors by 24 weeks after Blg-Cre activation? If these mice are followed out longer than 24 weeks, is there evidence that they would

eventually develop tumors? Do you have a hypothesis for why your Myc only model behaves differently than these other models?

Thank you for this comment; this is indeed an important point that we have now clarified in the revised manuscript. Specifically, we discuss the relative weakness of the *Rosa26* promoter and the fact that this is a knockin with only 2 copies of the *Myc* knockin gene, resulting in a level of MYC expression lower than most other transgenic *Myc* models; closer to the physiological level of MYC following mitogen stimulation, and as we have shown previously, 2-fold above control mammary gland levels of MYC (see Wang et al., 2011 [1-3]). In this setting of deregulated (constitutive), modest upregulation of MYC, like in human cancer, additional oncogenic hits are required to drive tumorigenesis. Even in a setting of *MYC* amplification (see manuscript Figure 1B), the *MYC* mRNA is only upregulated an average of 2-fold in human cancer; similar to our model of MYC expression. Indeed, although MYC is one of the most activated oncogenes implicated in the pathogenesis of human cancers, it has been shown that its activation alone generally results in the activation of checkpoints including those through p53, ARF, BIM, and PTEN that can cause cell growth arrest or death [4-7]. Thus, MYC cooperates with many other oncogenic or tumor suppressor genes to initiate tumorigenesis, as we are modeling here. Its activation is also generally essential for tumorigenesis as shown in several animal models of cancer, where MYC alteration is required for tumor initiation, progression, or maintenance [5, 8]. Therefore, tumors with dysregulated MYC have been considered as “MYC-driven” or “MYC-addicted” tumors. We have also previously published monitoring tumorigenesis with different Cre drivers in the ROSA-LSL-*Myc* model for 54 weeks with no tumors initiated [1-3].

2. The y-axis for Figure 1F goes from 0-150%. I would suggest changing this figure, so the y-axis is 0-100%.

Thank you for this suggestion. The Figure has been updated as you suggest.

3. In Figure 2F, the authors claim that “mice with SR tumors were sensitive to Paclitaxel”. To support this claim, the authors should show untreated tumor growth curves for both SR and SP tumors and perform a statistical analysis to show treatment with Paclitaxel reduces tumor growth.

Thank you for your comment. The controls were inadvertently left out. The correct figure is now updated with controls, statistics, and increased N number of Paclitaxel treated mice.

4. From the methods, it appears that SR and SP groupings for Figure 2F were determined post-treatment. Is it possible that treatment with Paclitaxel influences these groupings? For example, is it possible that Paclitaxel responsive SP tumors appear SR post-treatment and are incorrectly grouped?

Although SR and SP groupings for Figure 2F (now 2H) were scored post-treatment, the differences in tumor onset timing (Figure 2F) and the initial growth rate of SP and SR tumors (Figure 2G) is distinguishable prior to enrollment into treatment or control arms, and these were consistent with the endpoint histologies. Additionally, the ratios of SP and SR tumors determined post therapy are similar to the ratios in the control treated mice, and in the model in general. Thus, while we don't know what the histology of these tumors was before treatment, we anticipate that it has not changed, and we now discuss this point in the manuscript. We had considered doing a pre-therapy biopsy, and experimented with this, but thus far the biopsy quality is insufficient for histology and raises immune stimulation issues with the biopsy.

Reviewer #2, expertise in cyclic immunofluorescence and multiplex immunofluorescence (Remarks to the Author):

In this work, Doha et al. developed a mouse model of triple-negative breast cancer (TNBC) by mimicking two common TNBC mutations. A number of molecular experiments, including single-cell RNA sequencing and multiplexed immunohistochemistry, were conducted by the authors to demonstrate that this genetic mouse model replicates the molecular characteristics of human TNBC subtypes. This reviewer believes that this model would be extremely valuable for testing newly developed therapies. Overall, the manuscript is well organized, and the analysis is well supported by statistical data. I recommend the publication of this manuscript after addressing the following points.

1. Although the authors have already explored this topic in some detail in the publication, it would be helpful for future researchers who plan to use this model to provide a concise overview of how it differs from human TNBC. What are the limiting factors of the model? Which human TNBC molecular characteristics or subtypes were found to be missing from this model?

Thank you for this important suggestion. We have updated the manuscript discussion to include the following points describing caveats related to this model:

One: the correlation analysis performed between the model and human TNBC subtype signatures showed that only 1 of 13 tumors used for bulk RNA-seq showed a correlation to the Luminal Androgen Receptor (LAR) subtype, and there is less than 1% AR positive staining in the SP and SR tumor types. Thus, due to

low representation, the MycPten;fl model would be an inefficient model to study the LAR subtype of TNBC. However, the LAR subtype occurs in a significantly smaller number of TNBC cases [9]. Furthermore, it is known that androgen receptor (AR) expression is detected with the highest frequency in ER-positive breast cancer, where in TNBC, AR-positive tumors are considerably less common [10].

Two: Because the Blg-cre activation of Myc expression and deletion of Pten requires lactation (expression of Blg), the MycPten;fl model only develops tumors in female mice. While females represent >99% of breast cancer patients in humans [Qavi et al., 2021], this does represent an important caveat when using the MycPten;fl model to evaluate male TNBC and any systems or therapeutics which are dependent on endogenous levels of male associated androgens or other hormones.

Considering the significance of multiplexed imaging in this work, the corresponding methods portion is relatively concise. The authors stated that multiplexed imaging was performed by following the method described by Eng et al. (Communications Biology 5, 438, 2022).

2. Eng et al. reported that surgically sectioned tissue exhibits more cell loss during the CyCIF process. Even after five rounds, tissue prepared by surgical section had a cell loss of around 10%. Please indicate how the mouse tumor tissue was obtained as well as the cell loss percentage.

Thank you for your comment, the whole tumor was collected when mice were euthanized. Tumors were formalin-fixed and embedded in paraffin and two, 1.5 mm cores were punched in each tumor block to make the mouse TMA.

Within the mouse TMA, we segmented 760318 cells in round 1. After eight rounds of cycIF, 743672 cells remained and stained positive for DAPI in the nucleus. This represents a total cell loss of 2.2% after eight rounds of staining. We added the following text to the Methods: “Two percent of cells were filtered from analysis due to tissue loss (i.e. they were negative for DAPI staining after eight rounds of staining).”

3. The reviewer is impressed by the resemblance between the human expression data acquired by MIBI and the mouse model data acquired by CyCIF. These two imaging methods have distinct advantages and disadvantages. CyCIF images may include a high level of autofluorescence, whereas MIBI images do not. Eng et al. demonstrated that the level of autofluorescence in the AF488 channel varies based on various parameters and the way in which it may be removed theoretically and physically. For potential readers, it

would be beneficial to specify how autofluorescence was eliminated in each round and channel in the CyCIF imaging.

Thank you for the suggestion. We physically removed autofluorescence as in Eng et al. by using a pre-quenching step. We computationally removed autofluorescence (AF) using the scaled AF subtraction method from Eng et al., using R0 and R5Q autofluorescence images to produce scaled background images for subtraction. To clarify our AF subtraction process, we added the following text to the Methods:

“The mplexable image processing pipeline parameters are available as a jupyter notebook at: https://github.com/engjen/MYC-PTENfl-mouse/blob/main/20210707_RS-mTMA-20220119_mpleximage_data_extraction_pipeline_bue.ipynb

As specified in the pipeline notebook, we used a scaled autofluorescence subtraction strategy, using round zero (R0) autofluorescence images and round five quenched (R5Q) autofluorescence images to linearly interpolate and subtract autofluorescence that is changing over rounds of CyCIF (see ref 65).”

4. As noted by Eng et al., the order of antibodies influences their observed signal intensity. This reviewer is impressed by the similarities between the human- and mouse-related data depicted in Figure 5. Researchers who wish to conduct similar work will benefit from details on how the antibody orders were chosen and how the signal-to-noise ratio (SBR) measurement was validated.

We thank the reviewer for their suggestions and have made the following changes:

- Regarding antibody order, the following text was added to the Methods:

“Antibody order was chosen to minimize known artifacts including channel cross talk and incomplete quenching (see Supplemental Dataset 4). Specifically, antibodies that stain the same cell in the same cellular compartment were positioned in non-adjacent rounds and channels.”

- Regarding image QC and validation of SBR, the following text was added to the Methods:

“Each marker was validated for specificity and signal-to-background ratio (SBR) through visual review of the images. Twenty markers showing good specificity and SBR were selected for downstream analysis including cluster analysis.”

5. How were the dynamic ranges of the protein expression levels calculated when comparing the MIBI and CyCIF images? These two techniques have distinct dynamic ranges. Please explain how the absolute values of the protein expression levels depicted in figures 5g, 5h, 5j, and 5k were assessed and extracted from the raw pictures.

- Regarding feature extraction, we used the mplexable pipeline from Eng et al. (ref 65 in the main text) (<https://gitlab.com/engje/mplexable>), which is free and open source, so interested readers could reference the source code to find the method of extraction of intensity values from the raw images. See <https://gitlab.com/engje/mplexable/-/blob/master/mplexable/feat.py> line 76: “we use the scikit image function `measure.regionprops_table` to extract mean intensity, nuclear shape, and nuclear size”.
- For CyCIF, the values represent mean single cell pixel intensity values after AF subtraction on a 16-bit imaging system (i.e. theoretical range of 0 to 65K for each pixel; actual range of mean cellular intensities in our data is 0-8000 because the exposure time is chosen to not saturate images and the values plotted are the mean intensity [total intensity over area] so no single cell averages near the maximum pixel value).
- The MIBI images were segmented and single cell intensity was extracted by Patwa et al. of ref 66. The values represent units related to the time-of-flight mass spectrometry detector reported by Patwa et al..
- To clarify this, we added the following to the Methods:
 - For CyCIF, “We used the free and open-source mplexable pipeline for image registration, quality control, single cell segmentation and feature extraction.” And, “We extracted the pixel intensity values from autofluorescence subtracted 16-bit images using nucleus and cytoplasm segmentation masks to calculate the mean pixel intensity of each single subcellular compartment. For each marker, we selected the nuclear or cytoplasmic single cell mean intensity for analysis depending on expected intracellular distribution of that marker.”
 - For MIBI, “mean intensity values are based on time-of-flight mass spectrometry as reported in Patwa et al. of ref [66]

6. In multiple analyses, the authors used different resolutions in the Leiden algorithms for the unsupervised clustering. Please specify how the resolution was determined for such clustering would be helpful for readers who would like to replicate this work.

Thank you for this suggestion.

- We added the following text to the Methods to clarify our process for evaluation of clustering results for clustering of single cells in the CyCIF and MIBI datasets: “We tested three resolution parameters for the Leiden algorithm (0.4, 0.5 and 0.6). We performed clustering and visualized each cell type cluster’s spatial distribution in the tissue using the jupyter notebook available here: https://github.com/engjen/MYC-PTENfl-mouse/blob/main/20211027_RS-TMA_cluster.ipynb. We then visualized the spatial distribution of markers in the images using the napari image viewer to load the multichannel images available here: <https://www.synapse.org/#!Synapse:syn51314365/files/> and the code available here: https://github.com/engjen/MYC-PTENfl-mouse/blob/main/20211022_RS-TMAs_napari.py. The final resolution (0.6), resulting in 20 cell types, was selected based on visual examination of clustering results overlaid on images. Unsupervised clustering of cells in the MIBI dataset was performed with the Leiden algorithm using the same neighbor and resolution parameters as with the CyCIF dataset (neighbors 15, resolution = 0.6) resulting in 24 cell types. The code for clustering of MIBI celltypes, including visualization of each clusters’ spatial distribution in the tissue, is available here: https://github.com/engjen/MYC-PTENfl-mouse/blob/main/20211109_MIBI_cluster.ipynb.”
- We added the following text to the Methods to clarify our process for evaluation of clustering of tissues based on cell type frequency: “We tested four resolution parameters of Leiden algorithm (0.2, 0.3, 0.4 and 0.5) to cluster tissues based on cell frequency. We chose resolution 0.5, resulting in 6 clusters, because this captured the histological subtypes. Specifically, cluster 5, the highest in epithelial cells (or stroma poor+, SP+), was significantly correlated with the stroma poor histological subtype based on H&E evaluation (Chi-squared $p=2.2e-6$). We collapsed the 6 clusters [below - left] to three [below – right] based on correlation between clusters after hierarchical clustering (Supplementary Figure 8D).

Reviewer #3, expertise in single cell RNA sequencing (Remarks to the Author):

This manuscript by Doha et al. developed a TNBC murine model and performed bioinformatic analysis on these Myc;Pten^{fl} tumors to profile human TNBC. This study of TNBC could provide a valuable tool for the field to understand the microenvironment in TNBC patients and might have potential clinical significance.

However, a few aspects of the analysis might be improved to strengthen this manuscript and clarify the narrative.

1. In the RNA-seq analysis, please consider moving the PCA plot to the main figure to show the distinctive clustering between SR and SP samples. Then, in both Supp Figure 2D and main Figure 2H,I, it seems that some SP sample clusters with SR. Maybe the authors would want to elaborate on the nature of the outlier sample due to the relatively small sample size for each histology. In the heatmap, instead of showing all the genes without labeling, it might be clearer to just show the top genes in the principle components and perform a GO analysis using those genes.

We thank the reviewer for their suggestions and have made the following changes:

1. The original PCA plot was constructed using TPM normalized data, which Zhao et al (PMID: 34158060) identified as less reliable than normalized counts for grouping replicate samples. To ensure our results were not biased by these same technical artifacts, we have reprocessed the bulk RNA-seq data using normalized counts with DESeq2 and regenerated the plots (see Figure 2D-E). The manuscript, captions, methods, and code have all been updated accordingly.
2. The bulk RNA-seq PCA plot has been updated to visualize the first two principal components computed on the top 1000 variable genes after variance stabilizing transformation of the counts. This updated PCA plot is now included as Figure 2D.
3. We have updated the manuscript main text to include our interpretation of outliers in PC1 and PC2, and how they relate to established TNBC subtypes as follows: “Overall, the bulk transcriptional analysis suggests that the stromal poor samples are associated with tightly regulated transcriptional state, whereas the stromal rich samples demonstrate more heterogeneity in both principal component and gene space.” “The SR assigned MycPten;^{fl} tumor (b11) was both an outlier for correlation to the LAR subtype (spearman correlation = 0.3, mean across all samples = -0.02) and PC2 embedding, suggesting that genes with negative weights in PC2 may be related to the LAR TNBC subtype.”

4. We have updated the bulk RNA-seq heatmap Figure 2E to annotate whether each gene in the top 1000 variable genes has a positive or negative weight for PC1. We have additionally performed Gene Set Enrichment using PC1 and PC2 gene weights and found the results largely consistent with GSEA performed using differentially expressed genes between SP and SR subtypes (Figure 2I, Supplemental Figure S3A). To keep the manuscript focused and avoid repetitive results, particularly for the bulk RNAseq data when the single-cell analyses are more informative, we did not include this data in the revision, but have included it here for the reviewers. PC1 positive weighted genes (enriched in SP) are associated with altered metabolism via ion transport and PC1 negatively weighted genes (enriched in SR) are associated with cell motility and angiogenesis. PC2 positively weighted genes are associated with epithelial differentiation, and PC2 negatively weighted genes are associated with regulation of cell size.

2. In Supp Figure 7A, it is shown that Ecad is only expressed in the cluster on the right. If Ecad is used for manual gating of epithelial cells, why are epithelial cells also seen overlapping with stromal or immune cells in Supp Figure 7B? Please comment on the UMAP clustering patterns seen in Supp Figure 7B. The batch effect removal seems to be overdone such that the distinction between the three major populations is now missing.

Thank you for this comment. Ecad was not expressed in all tumor cell populations. Three cores contribute to the epithelial cells overlapping with stromal and immune cells (I11, G09 and H11)—since these cores are negative for Ecad, EpCAM and CK5 they end up on the left side of the UMAP near the stromal cell populations. However, you can see that they separate even farther to the left due to low eccentricity, larger

area and increased pMYC staining characteristic of epithelial tumor cells (and they were negative for all stromal and immune gating markers). Our methods describe the gating as follows:

“Endothelial cells were defined as CD31+. Epithelial cells were positive for 1 or more of Ecad, EpCAM and CK5 and CD31-. Immune cells were CD45+ CD31- and epithelial marker negative. Stromal cells were all non-endothelial, non-epithelial, non-immune segmented nuclei. Three cores had pMYC positive tumor cells negative for Ecad, EpCAM and CK5 (I11, G09 and H11). In these tissues, tumor cells were defined as any cells negative for all stromal and immune gating markers (CD31, CD45, Vim, aSMA, ColIV, ColIV, Gal3).”

However, we realize that our original S7B figure labelling is misleading. Therefore, we removed the misleading gating labels (left) and added a note: “** See methods for gating strategy” (right)

3. For Figure 6, to better illustrate the difference in cell composition between SR and SP, maybe the authors should consider showing a UMAP by the two phenotypes.

We thank the reviewer for their suggestion and have added a UMAP plot split by assignment of either Stromal Poor (SP), Stromal Rich Immune Poor (SR-IP) or Stromal Rich Immune Rich (SR-IR) tumors as Supplemental Figure S13F. This supplemental figure is now referenced in the main text. We also updated Figure 6B from a heatmap of cell counts per lineage to a bar plot, to ease visual interpretation of lineage distributions.

4. Most of Supp Figure S8 panels are not referenced in the manuscript, please fix. For a small dataset as is described in Supp Figure S8, please provide more rationale for clustering resolution. Panel 3 seems to be overclustering.

We have corrected this and all Figure S8 panels are now referenced in the manuscript.

As described in our response to reviewer 2, point 6, we have clarified our selection of clustering resolution in the revised manuscript. In Supplemental Figure S8, panel C, we chose resolution 0.5, resulting in 6 clusters, because this captured the histological subtypes. Specifically, cluster 5, the highest in epithelial cells (or stromal poor+, SP+), was significantly correlated with the stromal poor histological subtype based on H&E evaluation (Chi-squared $p=2.2e-6$). As you noted, clustering at the relatively high resolution necessary to separate out the SP+ group was over clustering, therefore, we collapsed the 6 clusters [see figure under response to reviewer 2, point 6 - left] to three [right] based on correlation between clusters after hierarchical clustering (Supplementary Figure S8D). This has been clarified in the Methods.

5. Please adjust the layout of Figure 5, some panels are overlapping.

Thank you for catching this, we have adjusted the layout of Figure 5 to ensure all elements are clearly displayed.

Other changes:

- We have made the text more concise to conform to Nature Communications word limit.
- We have added supplemental Figure S11C demonstrating the ability of FVB strain cells to be hash-tag labeled using the TotalSeq antibody kit, as it had not been reported to work in this strain and we have received a question about it from the community after posting a pre-print of this manuscript on bioRxiv.

We would like to again thank the reviewers for their helpful suggestions, which have allowed us to strengthen our manuscript. We understand the time required to review these manuscripts and greatly appreciate the reviewer's time and effort.

Response references:

1. Wang, X., et al., *Phosphorylation regulates c-Myc's oncogenic activity in the mammary gland*. Cancer Res, 2011. **71**(3): p. 925-36.
2. Daniel, C.J., et al., *T-cell Dysfunction upon Expression of MYC with Altered Phosphorylation at Threonine 58 and Serine 62*. Mol Cancer Res, 2022. **20**(7): p. 1151-1165.
3. Risom, T., et al., *Deregulating MYC in a model of HER2+ breast cancer mimics human intertumoral heterogeneity*. J Clin Invest, 2020. **130**(1): p. 231-246.
4. Evan, G.I., et al., *Induction of apoptosis in fibroblasts by c-myc protein*. Cell, 1992. **69**(1): p. 119-128.
5. Gabay, M., Y. Li, and D.W. Felsher, *MYC activation is a hallmark of cancer initiation and maintenance*. Cold Spring Harb Perspect Med, 2014. **4**(6).

6. Nilsson, J.A. and J.L. Cleveland, *Myc pathways provoking cell suicide and cancer*. *Oncogene*, 2003. **22**(56): p. 9007-21.
7. Murphy, D.J., et al., *Distinct thresholds govern Myc's biological output in vivo*. *Cancer Cell*, 2008. **14**(6): p. 447-57.
8. Vita, M. and M. Henriksson, *The Myc oncoprotein as a therapeutic target for human cancer*. *Seminars in Cancer Biology*, 2006. **16**(4): p. 318-330.
9. Le Du, F., et al., *Is the future of personalized therapy in triple-negative breast cancer based on molecular subtype?* *Oncotarget*, 2015. **6**(15): p. 12890.
10. Vidula, N., et al., *Androgen receptor gene expression in primary breast cancer*. *NPJ breast cancer*, 2019. **5**(1): p. 47.

Reviewers' Comments:

Reviewer #1:

Remarks to the Author:

Doha et al have provided a very thorough point-by-point response to all of the reviewer comments. Thank you. Based on their edits, I recommend this manuscript for publication.

Reviewer #2:

Remarks to the Author:

All of my comments have been addressed by the authors.

Reviewer #3:

Remarks to the Author:

The authors have adequately addressed all the previous comments. In the revised manuscript, the overall clarity and data presentation have been greatly improved.